# (Integration of automatic implicit geological modelling in deterministic geophysical inversion

Jérémie Giraud[1,2], Guillaume Caumon[1], Lachlan Grose[3], Vitaliy Ogarko[2,4] and Paul Cupillard[1]

[1] Université de Lorraine, CNRS, GeoRessources, F-54000 Nancy, France.
[2] Centre for Exploration Targeting (School of Earth Sciences), University of Western Australia, 35 Stirling Highway, 6009 Crawley, WA, Australia
[3] School of Earth Atmosphere and Environment, Monash University, Melbourne 3800, Australia
[4] Mineral Exploration Cooperative Research Centre, The University of Western Australia, 35 Stirling Highway, 6009 Crawley, WA, Australia

*Correspondence to*: Jeremie Giraud (jeremie.giraud@univ-lorraine.fr)

Abstract.

We propose and evaluate methods for the integration of automatic implicit geological modelling into the geophysical (potential field) inversion process. The objective is to enforce structural geological realism and to consider geological observations in level-set inversion, which inverts for the location of boundaries between rock units. We propose two approaches. In the first approach, a geological correction term is applied at each iteration of the inversion to reduce geological inconsistencies. This is achieved by integrating an automatic implicit geological modelling scheme within the geophysical inversion process. In the second approach, we use automatic geological modelling to derive a dynamic prior model term at each iteration of the inversion to limit departures from geologically feasible outcomes. We introduce the main theoretical aspects of the inversion algorithm and perform the proof-of-concept using two synthetic studies. The analysis of results using indicators measuring geophysical, petrophysical and structural geological misfits demonstrates that our approach effectively steers inversion towards geologically consistent models and reduces the risk of geologically unrealistic outcomes. Results suggest that geological correction may be effectively applied to pre-existing geophysical models to increase their geological realism and that it can also be used to explore geophysically equivalent models.

## 1. Introduction

One of the longstanding challenges faced by geophysical inversion in general, and potential field studies in particular, is the recovery of geologically meaningful inverse models. One of the chief factors explaining this is the strong non-uniqueness of the solution to the inverse problem: an infinite number of models can fit a given potential field dataset, including a vast space of geologically unrealistic outcomes. This has prompted the development of a number of approaches using prior information or constraints during inversion that aim at reducing the search space to models fitting the geophysical measurements (see, e.g., Lelièvre and Farquharson, 2016; Moorkamp, 2017; Wellmann and Caumon, 2018; Giraud et al., 2021b, and references therein). Earlier proposals comprise the use and design of regularisation schemes that account for prior information about the spatial variations of inverted properties (e.g., smoothness constraints, Li and Oldenburg , 1996) or departure from a reference model based on some hypothesis (e.g., smallness constraints, Hoerl and Kennard, 1970). A more recent, and drastic, approach to reduce the size of the search space is to consider the geometry of contact between rock units instead of the distribution of petrophysical values. In such case, the rock units' physical

properties are assumed to be known a priori and can be kept constant during inversion. This idea was proposed decades ago for ray-based inversion in reflection seismology (Gjoystdal et al., 1985), but it raises challenges to automatically maintain consistent relationships between geological interfaces (Caumon et al., 2004). More recently, surface-based inversion has become practical for seismic or for potential field inversion by using either an explicit formulation for the interface between rock units (Galley et al., 2020) or using level-sets in the inversion (e.g., Dahlke et al., 2020; Giraud et al., 2021a; Li et al., 2017, 2020; Rashidifard et al., 2021; Zheglova et al., 2018). In the implicit boundary representation, the unit boundaries correspond to the zero iso-value of the implicit functions representing the signed distance to interfaces. In this type of modelling, the algorithms invert directly for the location of the contacts between geological units by adjusting the location of these level-sets, allowing the automatic deformation of the geological units using geophysical data. Implicit formulations have a better ability than explicit surfaces to maintain volumetric validity throughout model updates and can additionally deal with topological changes (Collon et al., 2016; Wellmann and Caumon, 2018).

In exploration geophysics, recent studies have applied level-set inversion to the recovery of the geometry of one or two anomalous units, in both single physics or multi-physics inversion (Zheglova et al., 2018; Li et al., 2017, 2016). Subsequent works comprise the extension and modification of level-set inversion by Giraud et al. (2021a) and Rashidifard et al. (2021) whose framework addresses an arbitrary number of rock units in 3D gravity inversion. In comparison to the direct inversion of physical properties (i.e., density, electrical resistivity, seismic velocities, etc.), these geometrical inversions present a direct pathway to obtaining geologically realistic outcomes from the inversion of potential field data. Nonetheless, to the best of our knowledge, level-set inversions still lack the capability to ensure the geological plausibility of inverted models, in the sense that they can produce alterations of the original models that can potentially violate geological principles while exploring the geophysical data space. To mitigate this, several solutions may be devised. One possibility, explored recently by Güdük et al (2021) and Liang et al. (2023), consists in computing the geophysical response directly on geological models. In what follows, we propose two alternative approaches to integrate geophysical inversion and geological modelling that allow more freedom to the geophysical component of the workflow: to apply geological correction either during or after level-set based geophysical inversion. In the first case, one can think of ensuring geological plausibility a posteriori using an ad hoc process in which an existing geophysical inverse model undergoes modifications until it satisfies geological plausibility conditions. This could be applied, for instance, to existing rock unit models obtained from previous geophysical processing or interpretation. In the second case, there is the possibility to integrate geological modelling principles, data, and rules directly within the geophysical inversion algorithm. In this contribution, we will focus on, and explore, two avenues in this direction:

    a. The application of geological correction to the proposed model at each iteration of the geophysical inversion to ensure that the search for a model honouring the geophysical measurements does not decrease geological realism (introduced in sect. 3.3.1 and tested in sect. 5.1 and 5.2).

    b. Incorporating a geological term in the objective function of the geophysical inverse problem (introduced in section 3.3.2 and tested in section 5.2).

In the two points above, the recovery of geological parameters from models proposed during inversion is necessary. At each iteration of the inversion, geological quantities such as the orientation of a contact or its location are extracted from the current model and subsequently fed to a geological modelling engine. The geological modelling engine will, in turn propose the geological realisation closest to the geophysical inverse model, from which a 'geological correction' can be calculated and applied to the model update. This forms the basis for geological correction (point a. above) to ensure that

geological consistency with principles and data is maintained throughout inversion. The same principle is used in b., but to define a constraint term as part of the inversion's objective function.

The main object of this contribution is to introduce the methodology allowing the integration of automated geological modelling in the geophysical inversion process as mentioned above and to provide idealised proof-of-concepts in the form of two synthetic examples. This paper is articulated in six sections as follows. In the second section, we introduce the inversion algorithm that we use to integrate geological constraints. Following this, in sect. 3 we provide elements of implicit geological modelling required by the automated geological modelling process used to constrain geophysical

inversion. In this section, we also detail how geological constraints are applied using the approaches (a) and (b) mentioned above through an automated geological modelling process. In sect. 4, we introduce the series of metrics that we consider to assess inversion convergence and recovered models from both the geological and geophysical point of views. Sect. 5 presents the proof-of-concept using two synthetic examples of 3D models representing idealised scenarios. In the Discussion (Sect. 6), we place our findings in the broader context of subsurface modelling, discuss the limitations and

implications of our work and review potential extensions of the proposed method.

## 2. Geometrical inversion: formalization

### 2.1. Pre-requisite: Linking rock unit boundaries to physical property inversion

The proposed method relies on the formulation of the model using an implicit model formulation in the form of signed distances to interfaces between rock units (right-hand side of Fig. 1, right). As proposed by Giraud et al. (2021a), this

modelling approach considers 'rock units' as one or more rock types characterised by the same physical value (e.g., here, each unit is characterised by a single density value within the modelled area). Each rock unit is modelled by a unique signed-distance scalar field covering the study area. In a study considering $n_r$ rock units of known contrasting physical properties, we consider a set of $n_r$ signed-distance fields $\boldsymbol{\phi} = \{\boldsymbol{\phi}_k, k = 1, \dots, n_r\}$ over $n_m$ model cells corresponding to the distance to the boundaries of rock units. These signed-distances are calculated using the fast-marching method of

Sethian (1996) to maintain the following properties:

$$\boldsymbol{\phi}_k \begin{cases} > 0 \text{ inside unit k,} \\ = 0 \text{ at the boundary,} \\ < 0 \text{ outside unit k.} \end{cases} \tag{1}$$

In our level-set inversions, $\boldsymbol{\phi}$ is the primary variable inverted for, which constitutes the proxy for a direct link mapping geophysical and geological representations of the subsurface. We map these signed distances to petrophysical properties using:

$$\boldsymbol{m}(\boldsymbol{\phi}_1, \dots, \boldsymbol{\phi}_{n_r}) = \sum_{i=1}^{n_r} V_i H(\boldsymbol{\phi}_i) \left[ \prod_{j=1, j \neq i}^{n_r} \left(1 - H(\boldsymbol{\phi}_j)\right) \right], \tag{2}$$

where $\boldsymbol{V} \in \mathbb{R}^{n_r}$ is a vector storing the physical property value assigned to each of the $n_r$ geological units (e.g., density

contrasts for the different rock units in the case of gravity inversion). Similar to Giraud et al. (2021a), $H$ is the smeared-out Heaviside function, which we calculate following Osher and Fedkiw (2003). This smearing is useful for the calculation of the sensitivity matrix of the calculated data to changes in $\boldsymbol{\phi}$ from Eq. (2). Adapting it to our problem, $H$ is defined for the $k^{\text{th}}$ rock unit in the $i^{\text{th}}$ model-cell as

$$H(\phi_{k,i}) = \begin{cases} 0 \text{ if } \phi_{k,i} < -\tau_i, \\ \frac{1}{2} + \frac{\phi_{k,i}}{2\tau_i} + \frac{1}{2\pi}\sin\left(\frac{\pi\phi_{k,i}}{\tau_i}\right) \text{ if } 0 \leq |\phi_{k,i}| \leq \tau_i, \\ 1 \text{ if } \phi_{k,i} > \tau_i, \end{cases} \tag{3}$$

where $\boldsymbol{\tau} = \{\tau_i, i = 1, \dots, n_m\}$ defines the volume of rock where the boundary is allowed to vary between two successive iterations of the inversion. To the best of our knowledge, it is common to set all $\tau_i$ values to a constant, equal to $0.5 \times \min(\Delta x, \Delta y, \Delta z)$ (Li et al., 2017) in a regular mesh of cells with volume $\Delta x \times \Delta y \times \Delta z$. In our implementation, we extend this to the possibility to use spatially varying boundary thicknesses by allowing the neighbourhood defined by $\boldsymbol{\tau}$ to vary in space. This enables to anchor the model at observation points where $\boldsymbol{\tau}_i = 0$, e.g., at surface observations, boreholes, along seismic lines, etc. In extreme scenarios, the volume occupied by boundaries with $\tau_i \gg \Delta x, \Delta y, \Delta z$ of cells with volume $\Delta x \times \Delta y \times \Delta z$ may cover extensive parts of the study area, or, conversely, prevent the model from evolving when $\boldsymbol{\tau} = \mathbf{0}$ everywhere.

At each iteration of the inversion, $\boldsymbol{m}$ is updated from the changes in $\boldsymbol{\phi}$, $\delta\boldsymbol{\phi}$, which are required by the process of optimizing the objective function (also called cost function, see Section 2.2) within the domain defined by non-null values of $\boldsymbol{\tau}$.

## 2.2. General formulation

We formulate the inverse problem in the least-squares sense, taking gravity data inversion as an example. Adjusting the words of Giraud et al. (2021a), the choice of a least-squares framework is motivated by the flexibility it allows in the number of constraints and forms of prior information that can be used in the inversion. Note that it corresponds to the multi-Gaussian Bayesian inversion framework as a maximum a posteriori estimator (Tarantola, 2005).

The objective function to minimise reads:

$$\Psi(\boldsymbol{\phi}, \boldsymbol{d}^{obs}) = \|\boldsymbol{d}^{obs} - \boldsymbol{d}^{calc}\|_2^2 + \lambda_p\|\boldsymbol{W}_p(\boldsymbol{\phi} - \boldsymbol{\phi}^{prior})\|_2^2, \tag{4}$$

where $\boldsymbol{d}^{obs}$ are the observed data and $\boldsymbol{d}^{calc}$ the gravity response of the density contrast model $\boldsymbol{m}(\boldsymbol{\phi})$. We use $\boldsymbol{d}^{calc} = \boldsymbol{d}^{calc}(\boldsymbol{m}) = \boldsymbol{S}^m\boldsymbol{m}(\boldsymbol{\phi})$, where $\boldsymbol{S}^m$ is the sensitivity matrix of the gravity data $\boldsymbol{d}$ to changes in densities $\boldsymbol{m}(\boldsymbol{\phi})$. The first term of Eq. (4) corresponds to a data misfit term whereas the second term is a regularisation term that minimises deviations from the prior model; $\lambda_p$ is a positive scalar weighting the regularisation term; $\boldsymbol{W}_p$ is an inverse diagonal variance matrix of dimensions $(n_m n_r) \times (n_m n_r)$ whose values can vary in space according to prior information to favour or discourage specific changes or features in the model. We note that the definition of $\boldsymbol{W}_p$ used here differs from Giraud et al. (2021a), where $\boldsymbol{W}_p$ is constituted of line vectors of dimensions $n_r \times (n_m n_r)$. This allows for more flexibility to translate prior information into constraints on a cell by cell basis. $\boldsymbol{\phi}^{prior}$ is the signed distances of a prior model. For simplicity, we do not include data measurement and modelling errors in the term $\|\boldsymbol{d}^{obs} - \boldsymbol{d}^{calc}\|_2^2$ of Eq. (4), but instead stop the inversion when the solution reaches a prescribed misfit level. Naturally, variable measurement errors could be integrated by replacing this term by the more general expression $\|\boldsymbol{C}_d^{-1}(\boldsymbol{d}^{obs} - \boldsymbol{d}^{calc})\|_2^2$ where $\boldsymbol{C}_d$ would be the data covariance matrix.

We solve Eq. (4) iteratively and calculate the update of signed distances $\delta\boldsymbol{\phi}^k$ that reduces $\Psi(\boldsymbol{\phi}^k, \boldsymbol{d}^{obs})$ at each iteration $k$. To solve for $\delta\boldsymbol{\phi}^k$, we build the system of equations given in Appendix 1, which requires the sensitivity matrix $\boldsymbol{S}^{\phi,k}$ of $\boldsymbol{d}^{calc}(\boldsymbol{m}(\boldsymbol{\phi}^k))$ with respect to changes in $\boldsymbol{\phi}^k$ using the chain rule:

$$S^{\phi,k} = \frac{\partial \boldsymbol{d}^{calc}(\boldsymbol{m}(\boldsymbol{\phi}^k))}{\partial \boldsymbol{\phi}^k} = \frac{\partial \boldsymbol{d}}{\partial \boldsymbol{m}(\boldsymbol{\phi}^k)} \frac{\partial \boldsymbol{m}(\boldsymbol{\phi}^k)}{\partial \boldsymbol{\phi}^k} = \boldsymbol{S}^m \frac{\partial \boldsymbol{m}(\boldsymbol{\phi}^k)}{\partial \boldsymbol{\phi}^k}, \tag{5}$$

where $\frac{\partial \boldsymbol{m}(\boldsymbol{\phi}^k)}{\partial \boldsymbol{\phi}^k}$ is obtained analytically from Eqs. (2-3) (see Giraud et al., 2021a for details).

It can be shown that the objective function $\Psi(\boldsymbol{\phi}, \boldsymbol{d}^{obs})$ is equal to the log-posterior probability density distribution as formulated in the Bayesian framework (Tarantola 2005, Chapters 1 and 3 for more details). Here, the problem is therefore cast as a maximum a posteriori estimation. At the $k^{th}$ iteration, we calculate $\boldsymbol{\phi}^{k+1}$ such that $\boldsymbol{\phi}^{k+1} = \boldsymbol{\phi}^k + \delta\boldsymbol{\phi}^k$, and the updated model $\boldsymbol{m}(\boldsymbol{\phi}^{k+1})$ is calculated consistently with Eq. (2) by selecting the rock unit with the largest signed distance value at each model-cell $i = 1, \ldots, n_m$:

$$\boldsymbol{m}_i^{k+1} = \boldsymbol{V}_s \text{ where } s = \arg\max_s \left(\boldsymbol{\phi}_{s=1,\ldots,n_r}^{k+1}\right)_i. \tag{6}$$

We remind that the vector $\boldsymbol{V} \in \mathbb{R}^{n_r}$ contains the physical property values assigned to the different geological units (see Eq. 2). In Eq. (6), the *argmax* function leads to selecting the density contrast corresponding to the highest value of $\boldsymbol{\phi}$, which, intuitively, corresponds to the "innermost" rock unit. Following the same rationale as Zheglova et al. (2013), we then calculate the signed distances corresponding to the updated boundaries of $\boldsymbol{m}^{k+1}$ to maintain the signed distance properties of $\boldsymbol{\phi}^{k+1}$ as introduced in Sect. 2.1. We note that at any given iteration, the search space is restricted to the vicinity of boundaries between rock units as defined by the boundary's neighbourhood controlled by $\boldsymbol{\tau}$, which determines the current inversion's domain. This localisation dramatically reduces the volume of rock and the number of model-cells considered for modification between two successive iterations to satisfy geophysical data fit requirements.

**2.3. Prior model constraints on signed distances**

Using a prior structural geological model, the corresponding signed distances $\boldsymbol{\phi}^{prior}$ to boundaries can be calculated. We remind that in Eq. (4), the prior model constraints on signed distances are given as $\lambda_p \left\| \boldsymbol{W}_p (\boldsymbol{\phi} - \boldsymbol{\phi}^{prior}) \right\|_2^2$.

This allows the inversion to explore a part of the model space remaining within the neighbourhood of $\boldsymbol{\phi}^{prior}$. The size of this neighbourhood globally depends on $\lambda_p$, which controls the relative importance assigned to the prior model term during inversion compared to the geophysical data misfit term. It is also locally determined by $\boldsymbol{W}_p$, which tunes the importance of the prior model term for each model cell. Similar to other least-squares inversions, $\lambda_p$ can be set manually by trial and error, for example starting with a high value until model changes occur. Alternatively, the L-curve (Hansen and Johnston, 2001; Hansen and O'Leary, 1993) or general cross-validation principle (Farquharson and Oldenburg, 2004) may be used. In instances where a geological prior model is used, it is generally obtained before geophysical inversion and remains constant throughout inversion. In this contribution, our objective is to use as a prior model the result of a geological modelling process anchored only to the geological data and principles to better explore the admissible geological and geophysical parameter space.

## 3. Integrating structural geological modelling into geophysical inversion

The goal of this section is to introduce possible methods to extract geological information in the form of contact location and orientation data (angles) of geological features from inverted models that can be subsequently treated as geological data to implicit geological modelling.

### 3.1. Pre-requisite: implicit geological modelling in a nutshell

In implicit modelling, geological structures (e.g., faults, foliations, intrusions; and stratigraphic horizons) are represented by iso-values of one or several 3D scalar field (see Wellmann and Caumon, 2018, for a review). For example, fault surfaces are generated as iso-values of signed distance functions (possibly restricted to a given region of space), and strata as iso-values of a relative geological time function. For each geological surface or series of surfaces, the 3D scalar field is obtained by least-squares interpolation between spatial measurement points. In this paper, we use LoopStructural, which is an open-source Python library for implicit 3D geological modelling (Grose et al., 2021). In LoopStructural, geological features are modelled backwards in time starting with the most recent. Faults are modelled by first modelling the fault surface and fault displacement vector by building a structural frame consisting of three signed distance fields representing the fault geometry and kinematics. The fault can then be applied to the faulted features by restoring the observations of the faulted surface prior to interpolating the faulted surface. This means that the kinematics of the fault are directly incorporated into the surface description.

LoopStructural uses a discrete implicit modelling approach, where the implicit function is approximated using a piecewise combination of basis functions on a predefined support such as a linear tetrahedron on a tetrahedral mesh or a trilinear basis function on a Cartesian grid. Discrete implicit modelling forms an under-constrained system of equations because geological observations are sparse and there are usually more degrees of freedom than geological constraints (location or orientation of geological features). To ensure the stability of the solution, a continuous regularisation term is added. Usual choices for regularisation constraints are some type of discrete smoothness constraint (Frank et al., 2007; Irakarama et al., 2021) or minimisation of a continuous energy (Irakarama et al., 2022; Renaudeau et al., 2019). In this study, we use the finite difference regularisation as implemented in LoopStructural, which minimises the second derivative of the scalar fields in all directions using a finite difference scheme on a Cartesian grid, following Irakarama et al. (2021).

Geological observations such as the location of contacts, form lines, fault locations and structural measurements can constrain the value and/or the gradient of the implicit function (Frank et al., 2007). Geological observation (further denoted as $d^{geol}$) are incorporated by finding the mesh element which contains the observation point and adding the linear constraint for the relevant degrees of freedom (nodes of the element). Orientation data can be used to constrain the gradient of the implicit function $g$:

$$\nabla g(\mathbf{x}) = \mathbf{n}, \tag{7}$$

where $\mathbf{n}$ is the normal vector of the geological surface at the location $\mathbf{x}$. As Eq. (7) constrains both the direction and magnitude of the gradient of the scalar field, an alternative formulation to impose only the orientation is to find two tangent vectors, e.g., the strike vector $\mathbf{v}_{strike}$ and dip vector $\mathbf{v}_{dip}$, and to set

$$\begin{cases} \nabla g(\mathbf{x}) \cdot \mathbf{v}_{strike} = 0, \\ \nabla g(\mathbf{x}) \cdot \mathbf{v}_{dip} = 0. \end{cases} \tag{8}$$

Geological contacts or location of geological features are integrated into the implicit modelling by setting the value of the implicit function:

$$g(\mathbf{x}) = val, \qquad\qquad\qquad\qquad\qquad\qquad\qquad\qquad\qquad\qquad (9)$$

where $val$ is the value of the implicit function given at the location $\mathbf{x}$. The value should represent the distance to a reference horizon, for example 0 when the observation is located directly on the surface being modelled (e.g., a fault surface), or

the cumulative stratigraphic thickness to some reference horizon for different conforming stratigraphic interfaces. The implicit function is determined by solving the regularised, over determined problem using least squares minimisation. For this, LoopStructural uses a conjugate gradient algorithm to iteratively find the solution of the system of equations.

In the next subsections, we introduce how to recover input data for implicit geological modelling as mentioned above from models obtained through geophysical inversion. More specifically, we detail how to:

(1) Extract the location of contacts between units from geophysical regions to constrain stratigraphic contacts.

    (2) Retrieve orientation data from the plane approximating the location of contacts between non-conformable units to model an unconformity using both geological and geophysical modelling.

We note that while we use LoopStructural, the generation of geological models using the data provided in this paper can be carried out with other implicit geological modelling engines, such as through GeoModeller's application programming

interface (Calcagno et al., 2008; Guillen et al., 2008), GemPy (De La Varga et al., 2019), SKUA-GOCAD (Jayr et al., 2008), Petrel (Souche et al., 2015), or Leapfrog (Cowan and Beatson, 2002).

### 3.2. Recovering structural information from and for geophysical inversion

### 3.2.1. Stratigraphic information

In implicit geological modelling, interfaces are defined by iso-values of one or several scalar fields analogous to signed

distances or to relative geological time. This opens up pathways to integrate implicit geological modelling and level-set inversion as introduced above and illustrated in Figure 2**Erreur ! Source du renvoi introuvable.**. Stratigraphic information are recovered from the current geophysical model by identification of the contacts between rock units. More specifically, the 3D coordinates of the top of the different units within a given layered stratigraphy are extracted from 3D rock unit models and stored as input data (as in Eq. 9) for implicit geological modelling.

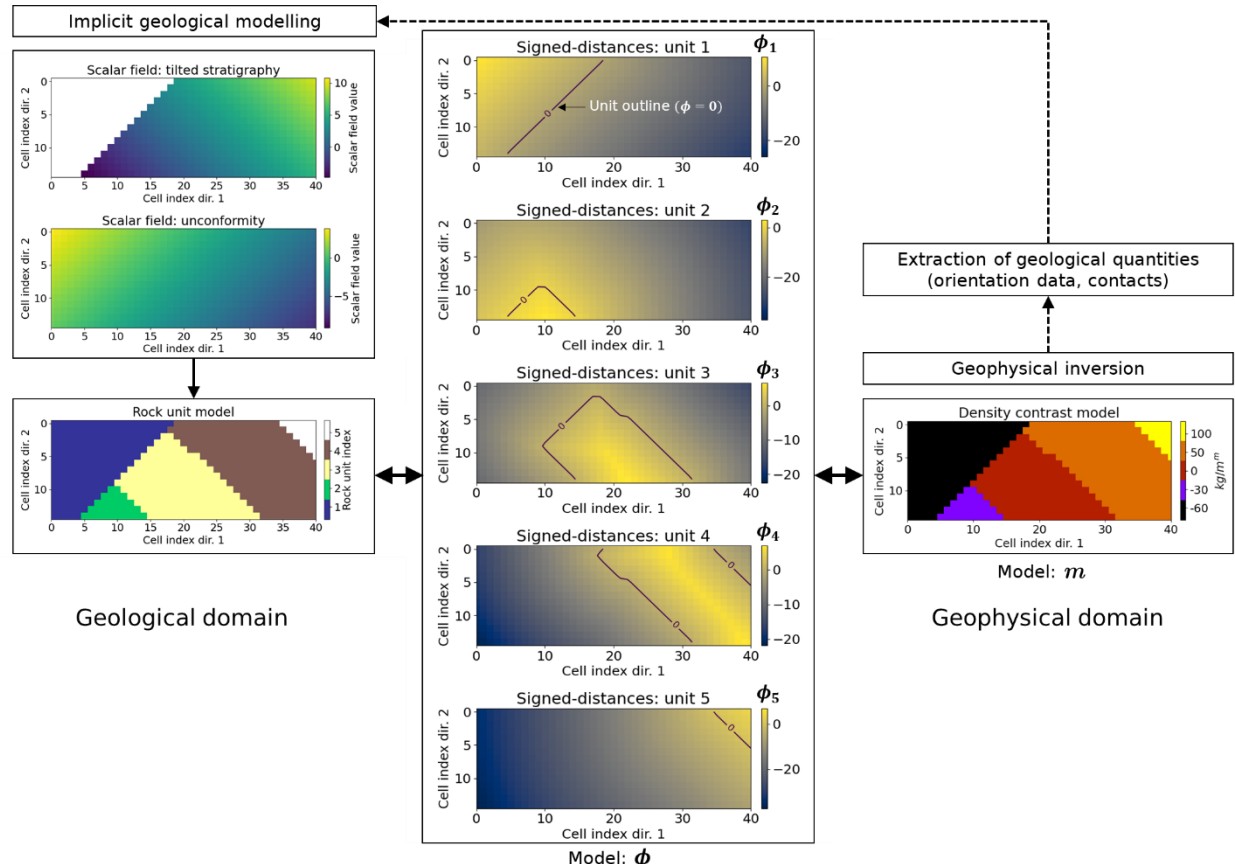

**Figure 1. Proposed strategy to link implicit geological modelling with geophysical level-set inversion. The geological domain represents implicit geological modelling (left hand side) and relates to Sect. 3.1 and 3.2. The middle and right hand side panels illustrates the link between signed distances and density contrasts introduced in Sect. 2.1. The dashed line arrow connecting geophysical inversion to geological domain symbolises the exchange of information between geological and geophysical modelling that can be used to link the two modelling processes.**

### 3.2.2. Orientation observations: example of a subplanar unconformity

In some cases, interpretive orientation data can be used as input to geological modelling (e.g., Sprague and de Kemp, 2005). Similarly, the geophysical level-set approach provides region boundary orientation which can be used to locally constrain the planar orientations of geological interfaces.

For instance, let us consider the example of a roughly planar erosion surface affecting some older stratigraphic series, which constitute an unconformity. We propose to recover the average normal vector **n** to this plane, which is required for the implicit model to be well posed in the absence of post-erosional stratigraphic data, by:

1)  Identification of the contact locations for the surface of interest from current geophysical model $\phi$, followed by

2)  Calculation of the best-fitting plane approximating the identified locations of the selected contacts (constrained least-squares fit through a least-squares minimization process, see Appendix 2).

After it is recovered, the vector **n** is normalised and used with Eqs. (7) or (8) in implicit geological modelling.

### 3.3. Automated geological modelling during inversion

#### 3.3.1. Geological correction

A way to promote geological consistency at each iteration of the geophysical inversion is to adjust the model update $\delta\boldsymbol{\phi}$ to limit changes in $\boldsymbol{\phi}$ that contradict geological data and principles. For this, we apply what we further refer to as a 'geological correction term' to the update term $\delta\boldsymbol{\phi}$ obtained from solving Eq. (4). In what follows, we introduce the 'geological' signed distances or relative geological time values $\boldsymbol{f}^{geol}$ to the rock units corresponding to the geological model derived from $\boldsymbol{d}^{geol}$ and the current geophysical update of $\boldsymbol{\phi}$. $\boldsymbol{f}^{geol}$ can be seen as a re-parameterisation of the geological model in a way that is compatible with the geophysical signed distances values $\boldsymbol{\phi}$. In other words, the application of $\boldsymbol{f}^{geol}$ consists in the calculation of the geological image of the geophysical signed distances values $\boldsymbol{\phi}$. It is computed using the following:

1) Extraction of geological information from the current signed distance model $\boldsymbol{\phi}_*^k$ (contacts and orientation data corresponding to the current model, see sect. 3.2.1),

2) Utilisation of this geological information as input to an implicit geological modelling engine (here, LoopStructural), where it is used to calculate the corresponding geological model together with geological data $\boldsymbol{d}^{geol}$, and

3) Computation of signed distances $\boldsymbol{f}^{geol}$ from the geological model to calculate the 'geological correction term' $\delta\boldsymbol{\phi}^{geol}$.

At the $k^{\text{th}}$ iteration, we first calculate $\boldsymbol{\phi}_*^k$, the updated signed distance obtained from solving the geophysical inverse problem formulated in Eq. (4) around the current model:

$$\boldsymbol{\phi}_*^k = \boldsymbol{\phi}^{k-1} + \delta\boldsymbol{\phi}^k, \tag{10}$$

and then we use it to calculate the geological correction term:

$$\delta\boldsymbol{\phi}^{geol}(\boldsymbol{\phi}_*^k, \boldsymbol{d}^{geol}) = \boldsymbol{f}^{geol}(\boldsymbol{\phi}_*^k, \boldsymbol{d}^{geol}) - \boldsymbol{\phi}_*^k. \tag{11}$$

Calculating $\boldsymbol{f}^{geol}(\boldsymbol{\phi}_*^k, \boldsymbol{d}^{geol})$ provides the closest geological model honouring geological information extracted from $\boldsymbol{\phi}_*^k$ together with geological data and knowledge encapsulated in $\boldsymbol{d}^{geol}$ (i.e., it is the geological image of image of $\boldsymbol{\phi}_*^k$). In this way, $\boldsymbol{f}^{geol}(\boldsymbol{\phi}_*^k, \boldsymbol{d}^{geol})$ returns a set of signed distance values which account for the relation between units (e.g., stratigraphic thickness, age relationships), known locations of contacts (e.g., seismic interpretation, borehole data, and surface geological observations), orientation data, and models proposed by geophysical inversion. Using $\delta\boldsymbol{\phi}^{geol}$, we update the signed distances as follows:

$$\boldsymbol{\phi}^k = \boldsymbol{\phi}_*^k + \alpha\delta\boldsymbol{\phi}^{geol}(\boldsymbol{\phi}_*^k, \boldsymbol{d}^{geol}) = (1-\alpha)\boldsymbol{\phi}_*^k + \alpha\boldsymbol{f}^{geol}(\boldsymbol{\phi}_*^k, \boldsymbol{d}^{geol}), \alpha \in [0,1[, \tag{12}$$

where $\alpha$ adjusts the importance given to the geological correction term.

As a consequence of Eq. (11), $\delta\boldsymbol{\phi}^{geol}(\boldsymbol{\phi}_*^k)$ is equal to 0 at all locations the proposed geophysical update $\delta\boldsymbol{\phi}$ does not conflict with geological modelling, and differs elsewhere, thereby steering the inversion towards the region of the geophysical model space corresponding to geologically consistent models. The contribution of the geological term to model update during geophysical inversion is illustrated in Figure 2 following ①.

In what follows, we set $\alpha = 1/2$ to balance the contributions of the different terms.

### 3.3.2. Geological term into the cost function

A possible shortcoming of the approach proposed in 3.3.1 is that the geophysical solution $\phi^k_*$ at iteration $k$ remains anchored on the prior model $\phi^{prior}$ (Eq. 4). In this section, we propose instead to integrate geological modelling in the cost function so that inversion can explore a larger portion of the model space. This can be achieved by considering the implicit geological model calculated in the same fashion as $\delta\phi^{geol}$ in Section 3.3.1. In such a case, $\delta\phi^{geol}$ can be used as a substitute for $\delta\phi^{prior}$ by setting $\delta\phi^{prior} = \phi - \phi^{prior} = \delta\phi^{geol}(\phi^k, d^{geol})$ in Eq. (4) to solve the problem at the next iteration (flow ② in Figure 2). Therefore, Eq. (4) becomes a function of $d^{geol}$ and $\Psi(\phi, d^{obs})$ rewrites as $\Psi(\phi, d^{obs}, d^{geol})$ as the inversion solves a geophysical and geological problem at each iteration. At iteration $k$, combining Eq. (4) and Eq. (12), we obtain the update of the signed distances by minimizing:

$$\Psi(\phi, d^{obs}, d^{geol}) = \|d^{obs} - d^{calc}\|^2_2 + \lambda_p \|W_p(\phi - f^{geol}(\phi^{k-1}, d^{geol}))\|^2_2, \tag{13}$$

where we set $\phi^{geol} = f^{geol}$.

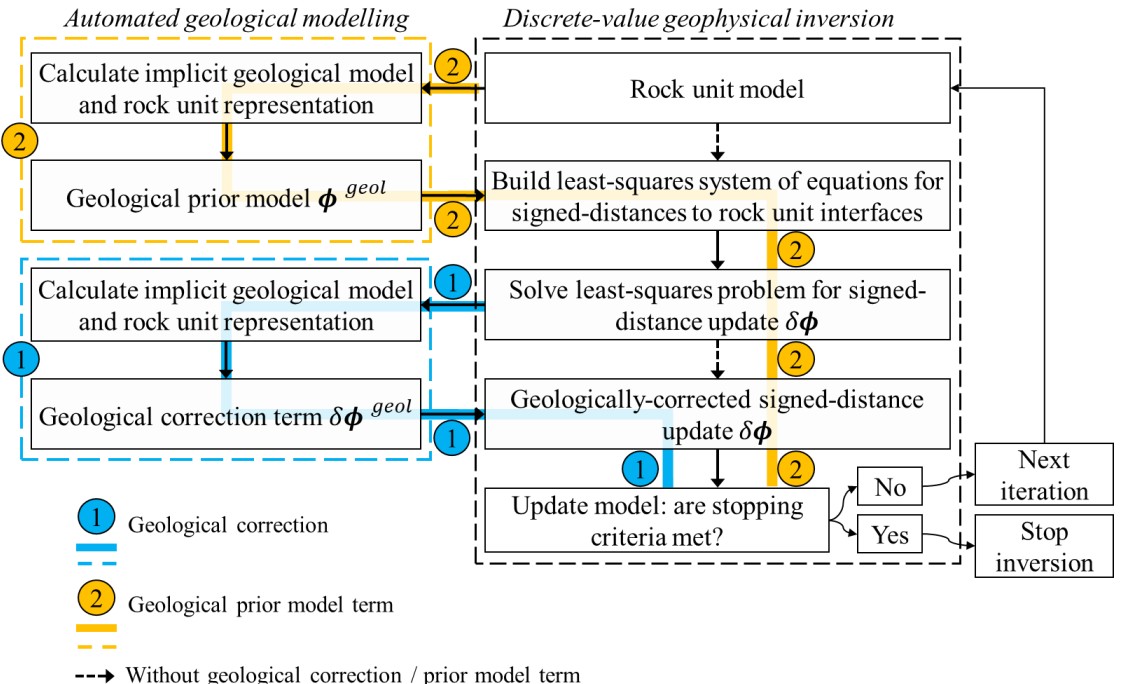

**Figure 2. Summary of the proposed approaches to integrated geological modelling in geophysical inversion: iterative use of the geological correction term with a fixed prior (1) or update of the prior geological model (2). Note that the combination of both is possible. Modified from Giraud et al. (2022).**

## 4. Metrics for the evaluation of inversion results

The above inverse methodologies can produce different outcomes for the inverted models in terms of the rock unit geometries and spatial distribution of physical properties. For the evaluation of inversion results, this section proposes metrics adapted to the chosen model parameterisation.

### 4.1. Overlap coefficient

The overlap coefficient ($OC$) (Szymkiewicz, 2017) is a similarity measure related to the Jaccard index (Jaccard, 1901) that measures the overlap between two sets. Applied to geological modelling, it is a measure of the dissimilarity between

discrete representations of the subsurface. For the comparison of the inverted rock type model $\boldsymbol{m}^{inv}$ and the reference model $\boldsymbol{m}^{ref}$, $OC$ can be written as:

$$OC(\boldsymbol{m}^{inv}, \boldsymbol{m}^{ref}) = \frac{\text{card}(\boldsymbol{m}^{ref} \cap \boldsymbol{m}^{inv})}{\min(\text{card}(\boldsymbol{m}^{ref}), \text{card}(\boldsymbol{m}^{inv}))} = \frac{\mathbf{1}}{\boldsymbol{n_m}} \sum_{i=1}^{n_m} \mathbb{1}_{m_i^{ref} = m_i^{inv}}, \tag{14}$$

where $\cap$ denotes the intersection of values of sets; card is the cardinality operator, which returns the size of a given set; $\text{card}(\boldsymbol{m}^{ref} \cap \boldsymbol{m}^{inv})$ is number of model cells where the rock types from $\boldsymbol{m}^{ref}$ and $\boldsymbol{m}^{inv}$ are the same, while

$\min(\text{card}(\boldsymbol{m}^{ref}), \text{card}(\boldsymbol{m}^{inv}))$ returns the size of the smallest set (here, the number of model-cells); $\mathbb{1}$ is the indicator function such that $\mathbb{1}_{m_i^{ref} = m_i^{inv}} = 1$ if $m_i^{ref}$ and $m_i^{inv}$ are equal, and $\mathbb{1}_{m_i^{ref} = m_i^{inv}} = 0$ otherwise. In this paper, $\boldsymbol{m}^{ref}$ and $\boldsymbol{m}^{inv}$ have the same discretization. Therefore, $OC$ represents the relative volume of rock assigned with the correct rock unit. Full dissimilarity is characterised by a value of $OC$ equal to 0 and perfect similarity is characterised by a value of 1.

### 4.2. Density contrast model misfit

In our analysis of synthetic cases, we assess the ability of inversion to recover the reference density contrast model using the root-mean-square error $ERR_m$ as a measure of the difference between the reference and inverted models:

$$ERR_m(\boldsymbol{m}^{ref}, \boldsymbol{m}^{inv}) = \sqrt{\frac{1}{n_m} \sum_{i}^{n_m} (m_i^{ref} - m_i^{inv})^2}, \tag{15}$$

It corresponds to the standard deviation of the misfit between retrieved and reference models. It is routinely used to evaluate the capacity of inversion algorithms to recover the reference petrophysical model in synthetic studies.

### 4.3. Geophysical data misfit

We assess whether the estimated model adequately reflects the measured geophysical data and monitor the inversion's stability using the root-mean-square error $ERR_d$ as a measure of the geophysical data misfit. It corresponds to a normalisation of the data misfit term in Eq. (4). We calculate it as:

$$ERR_d(\boldsymbol{m}^{inv}) = \sqrt{\frac{1}{n_d} \sum_{i}^{n_d} (d_i^{obs} - d_i^{calc})^2}. \tag{16}$$

It is one of the metrics most commonly used to evaluate the capability of inversion to reproduce field measurements.

### 4.4. Adjacency matrix

Similar to the posterior analysis of Giraud et al. (2019), we analyse rock unit models recovered from inversion using adjacency matrices. Adjacency defines which rock bodies are in contact (Egenhofer and Herring, 1990), and is one of the simplest ways to assess a geological model from a quantitative point of view. For details, we refer the reader to Pellerin et al. (2015) and Thiele et al. (2016), who show its usefulness in the context of geological modelling. In this work, we simply use the number of grid faces located at the boundary between units with indices $i$ and $j$, respectively, as coefficient

for $A_{i,j}$ of the corresponding $n_r \times n_r$ adjacency matrix $\boldsymbol{A}$. From a more abstract standpoint, this representation amounts to consider the geological model as a non-oriented graph (Godsil and Royle, 2001), where nodes correspond to the rock units and edges correspond to adjacency relationships. It can be calculated globally for a general overview (i.e., one

adjacency matrix calculated for the full model), or locally for more detailed analysis (i.e., adjacency matrices calculated only at certain locations).

### 4.5. Signed distances misfit

To quantify the difference between rock unit boundary locations in the reference and recovered models from an implicit modelling point of view, we propose a metric using signed distances to these interfaces:

$$ERR_\phi(\boldsymbol{\phi}^{ref}, \boldsymbol{\phi}^{inv}) = \sqrt{\frac{1}{n_r}\frac{1}{n_m}\sum_i^{n_r}\sum_j^{n_m}\left(\phi_{ij}^{ref} - \phi_{ij}^{inv}\right)^2}. \tag{17}$$

Like the density contrast model misfit measures $ERR_m$ introduced above, the signed distances misfit $ERR_\phi$ measures the discrepancy between two models. Here, it offers a quantitative insight into the distance between interfaces of two structural models with the same discretisation.

## 5. Synthetic application cases

This section introduces the proof of concept of the proposed approach using two idealised examples. They illustrate the capability of the proposed inversion scheme to interleave geological modelling and geophysical inversion to recover geologically consistent models. We first explore the case of a layered stratigraphy before moving on with an example of the investigation of the dip of a planar unconformity.

### 5.1. Geological correction: layered stratigraphy

#### 5.1.1. Survey setup

The synthetic example presented here is an extension from Giraud et al. (2022), which shows a summarised example of the use of a geological correction term. We present it in more details and expand on the analysis and interpretation of results.

The reference geological structural model is generated starting from the *Claudius* dataset in the Carnarvon Basin (Western Australia, interpreted from WesternGeco seismic data made available by Geoscience Australia). This real world dataset is freely available online for benchmarking purposes (https://github.com/Loop3D/ImplicitBenchmark, last accessed on 30/07/2023) and used as a toy model in the LoopStructural package (Grose et al. 2021a). Here, the *Claudius* dataset, which consists of points sampled from interpreted seismic horizons in 3D, is used for the generation of implicit models in LoopStructural.

In this work, we start from an upturned version of the original model (Figure 4a). We assume that two hypothetic perpendicular 2D seismic profiles (see their location in Figure 3c) together with general knowledge of the area provide sufficient information to build a prior rock model, from which $\boldsymbol{\phi}^{prior}$ is calculated. We assume that these seismic profiles and their close neighbourhood can be treated as low uncertainty zones. Low uncertainty areas also comprise the single shallowest layer of model-cells of the model under the assumption that the top layer can be well-constrained by geological field observations such as the nature of directly observable rocks. To convey increasing uncertainty with distance to seismic section, we assign $\boldsymbol{W}$ with values inversely proportional to the squared distance to the seismic profiles (Figure 3d), starting from a value of 1 along the profiles. As a consequence, the prior model weight (Figure 3d) decreases rapidly

with distance to the seismic lines. Inversion is, therefore, mostly free to update the model as $W_{ij} \ll 1$ in large a portion of the study area away from the seismic lines while remaining strongly influenced in their vicinity.

For our testing, we modified the original geological model further with the manual exaggeration of a dome present in the original model which affects all units in the synthetic example Figure 3a. The resulting model is shown Figure 4a, where it is marked by the red arrow at the intersection of the two vertical slices. It is characterised by a vertical Gaussian

displacement field with amplitude 500 m and standard deviation 350 m in both horizontal directions, centred around coordinates Easting = 2,700 m and Northing = 2,125 m. We note from Figure 3b that the added dome constitutes a noticeable difference with the starting model.

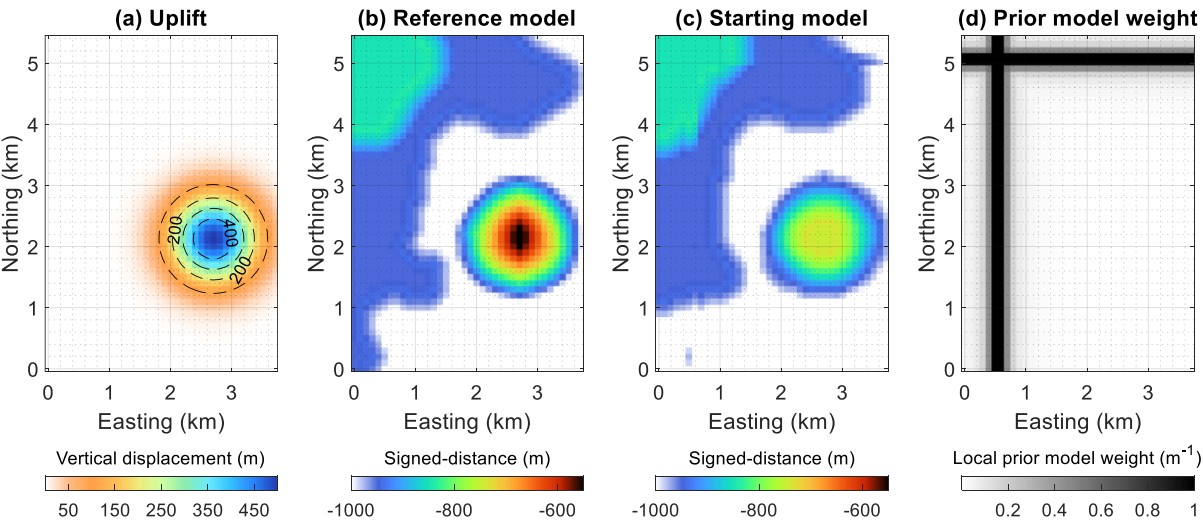

**Figure 3. (a) Dome added to the original model, (b) reference model: signed distance field of unit 1 at depth equal to -1000 meters, (c) starting model: signed distance field of unit 1 at depth equal to -1000 meters, (d) weights $W_p$ assigned to the prior model term in Eq. (4). We note that the starting model as shown in (c) also corresponds to the prior model.**

In what follows, we test the capability of level-set based inversion to recover the uplift, both without and with geological correction. We use a starting model where the dome is nearly missing (see Figure 3b for the example of Unit 1, and Figure

4b for a 3D view of the model), simulating the scenario in which little to no indication is present in the 2D seismic and geological information. In addition to the dome, we increased the discrepancy between the starting model and the reference model by subsampling the reference geological dataset generating the starting model. It is obtained by retaining one out of every nine points of the original dataset (i.e., points from 2D surfaces) used to generate the reference model in LoopStructural. This generates fine-scale variations of the model, as can be seen from the comparison of Figure 3b and

Figure 3c. From the comparison of the gravity data shown in Figure 4a and Figure 4b, it appears that the perturbations of the reference model generate a strong starting data misfit for the starting model. We set up inversions such that the starting model is equal to $\boldsymbol{\phi}^{start} = \boldsymbol{\phi}^{prior}$. To define the geological data $\boldsymbol{d}^{geol}$ used in the calculation of the geological correction term as in Eqs. (7-8), we assume only knowledge of the stratigraphic column and of the average orientation of layers. In this example, the stratigraphic column is conformable, meaning that all geological layers are represented with one

continuous relative geological time function. Consequently, the influence of geological correction should be to direct inversion towards a model of conformable layers arranged following the deposition order encoded in the stratigraphic column.

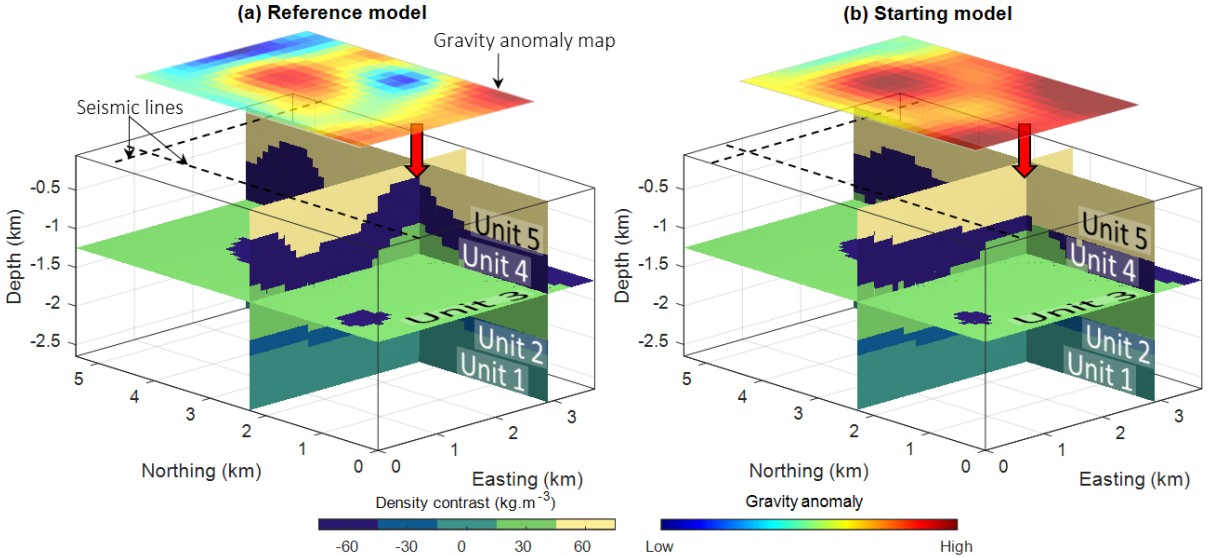

**Figure 4. Synthetic model for proof-of-concept testing: reference model (a) with the corresponding gravity anomaly shown in transparency and starting model (b). The location of the seismic sections used to derive the prior model is shown by the dashed lines. The red arrow shows the dome location. The two colour bars and their respective palettes are common to (a) and (b). Modified and adjusted from Giraud et al. (2022).**

### 5.1.2. Inversion results and interpretation

Due to the overall simplicity of the model, the inversion converges in about 10 iterations, taking only a few seconds on a laptop computer. Inversion results are shown in Figure 5.

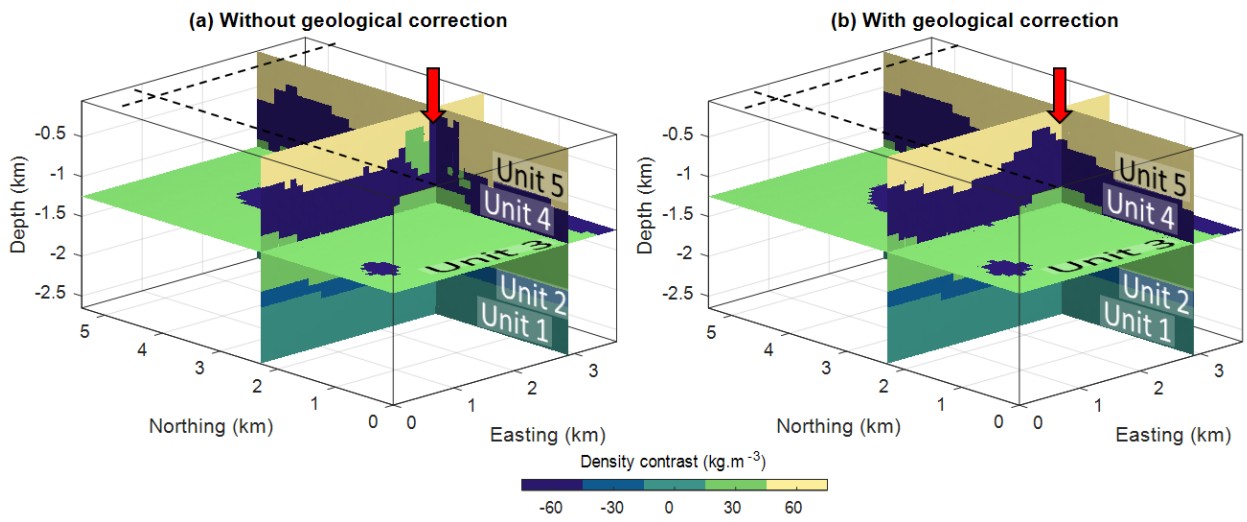

**Figure 5. Inversion results for proof-of-concept: without application of geological correction (a) and using geological correction (b). Modified and adjusted from Giraud et al. (2022).**

When no geological correction is applied ($\alpha = 0$ in Eq. 12), the requirement to reduce the data misfit component of Eq. (4) (the evolution of which is shown in Figure 4a and Figure 4b) leads the inversion to produce geologically unfeasible features (abnormal stratigraphic contacts, Figure 5a). On the contrary, consistent stratigraphic contacts and conformable stratigraphic units are obtained when the geological correction is applied with $\alpha = 0.5$ (Figure 5b). Visually, the recovered model looks comparable to the reference model. Because the two recovered models have a different rock type representation, we consider other indicators to obtain a finer analysis. The value of $\alpha = 0.5$ was chosen without a rigorous

analysis of its impact on the results. A naïve trial and error approach revealed that a value of 0.5 effectively 'corrected' the course taken by un-corrected geophysical inversion and prevented the appearance of artefacts.

We complement our comparison of inverted models using adjacency relationships introduced in Section 4.4. In a layered
stratigraphy such as presented here, this can be useful to identify geological contacts violating age relationships. In addition, it may be an indicator of the ruggedness of surface contact as it measures the overall contact area. To compare the recovered models with the reference model, we calculate the difference between their respective adjacency matrices (Figure 6). In Figure 6a and Figure 6b, we observe the occurrence of contacts absent from the reference model, where adjacency between units only follows the depositional order (the stratigraphic column) (see Figure 6d). Following
geological rules, the contacts between units 3 and 5 recorded by the adjacency matrices of the starting model (Figure 6a) and inversion without geological correction (Figure 6a) should be forbidden. The comparison of adjacency matrices indicates that, in this case, inversion allows contact between units that are in disagreement with the reference model and which violate geological principles. It is interesting to notice, however, that geophysical inversion reduces the number of such contacts even in the absence of geological correction.

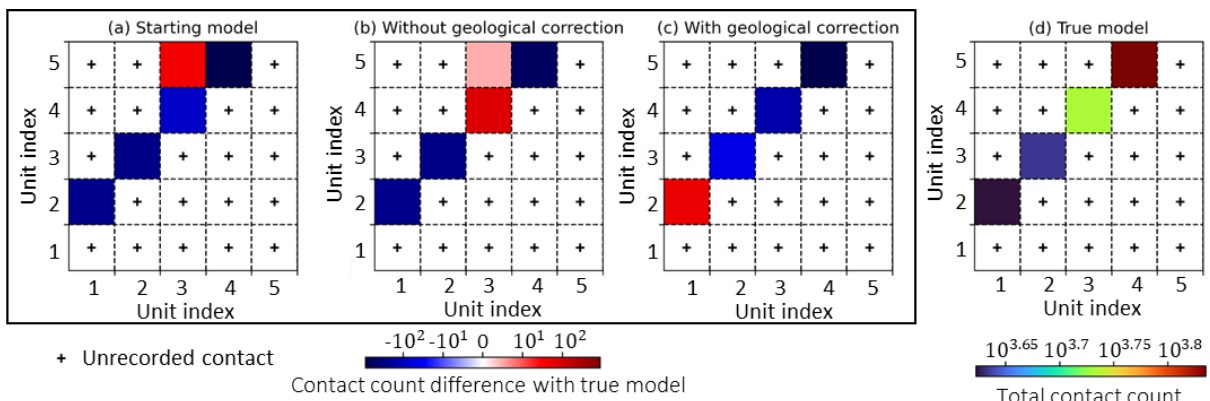

**Figure 6. Adjacency matrices: differences between the starting model (a), inverted model without geological correction (b), and with geological correction used during inversion (c); (d) shows the adjacency matrix of the true model. Adjacency relationships are represented using upper triangular adjacency matrices as adjacency relationships are symmetric. The diagonal is left empty because we do not record occurrences of a rock in contact with itself.**

Further, the geological correction term reduces the model search space to outcomes in agreement with the geological knowledge infused during inversion. While it is possible that such contacts come about at intermediate steps of the inversion, convergence of the algorithm makes it unlikely for them to persist.

From the success of this synthetic test, we have developed a structurally more complex model to investigate other features of the proposed algorithms and evaluate the limits of the integration method. While analysing the influence of inaccurate
knowledge of densities in detail is beyond the scope of this paper, it remains important to ensure that inversions are robust to small errors in density. For this, we refer the reader to Appendix 5 where we simulate errors in the knowledge of unit 4. Previous works using level inversion have investigated the importance of the starting model in the uncorrected case and assume that their conclusions hold. Likewise, we assume robustness of level set inversion to noise in the data as it was shown by previous works cited in Sect. 1. To confirm this and for completeness, we performed additional tests, using:
• Data contaminated with noise relatively high compared to the amplitude of the uncontaminated data (Appendix 3: Robustness to noise);

- A degenerate starting model and data contaminated with noise (Appendix 4: Robustness to a degenerate starting model)

- A starting model affected by errors in the density of rocks (Appendix 5: Robustness to errors in the density of rock units).

A detailed analysis of these tests is beyond the scope of this paper and we refer the reader to these Appendices for more detail. In the remainder of this article, we assume that our approach is sufficiently robust to random noise and inaccurate starting models.

**5.2. Testing the inversion approaches in the presence of an unconformity**

In this section, we investigate a more challenging geological setting and explore the possibility to use automatic geological modelling to define a term from the inversion's objective function. We also test the possibility to combine it with the application of geological correction to the model update. Additionally, we examine the possibility to use geological correction a posteriori to ensure geological realism (to 'geologify') an existing model presenting features that conflict

with geological principles and/or data.

**5.2.1. Survey setup**

We generate a synthetic model to test the proposed approach to recover information about objects others than conformable horizons such as unconformities. To this end, we generate a reference model resulting from three main geological events occurring in the following order (Fig. 8):

(1) Deposition of isopach stratigraphy made of 4 units and regional tilting.

(2) Sinistral faulting of these layers.

(3) Erosion followed by a new depositional episode, leading to the observation of an angular unconformity.

These geological features can be produced using implicit modelling as follows. As explained above, all units within a conformable stratigraphic unit can be modelled using the same scalar field, which can be assimilated to a signed distance

to some reference horizon. This signed distance represents conformable horizons where the value of the scalar field corresponds to the cumulative thickness from the base of the modelled series. The stratigraphic column defines the horizons based on these cumulative thicknesses. The rock units then correspond to thickness intervals which are associated to a rock model, and are associated to density contrasts (Fig. 8a). The orientation of parallel layers is governed by a vector indicating the direction towards younger strata (later referred to as the 'younging' direction). The unconformity is assumed

to be planar, so it is fully defined by a normal vector and a location point. This erosion surface separates two groups of stratigraphic units. For simplicity, we only consider a single rock unit overlying the unconformity (displayed in red in Fig. 8). The data used to generate the model is given in Appendix 6: Geological data, which provides quantities used in Eqs. (7)-(9). A view of the density contrast corresponding to this reference model is shown in Figure 8.

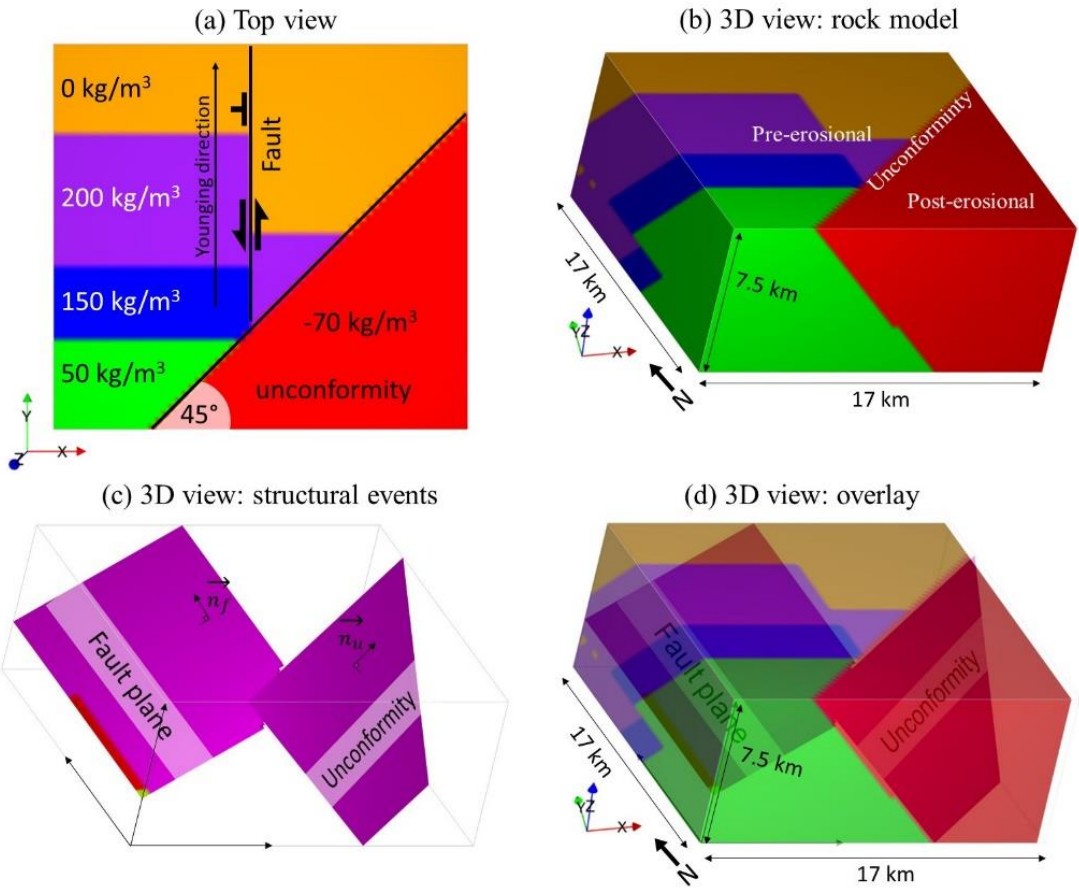

Figure 7. Reference geological model and density contrasts: (a) top view, (b) 3D rock unit cube and (c) structural events, overlayed in (d).

For our tests using this model, we consider a case where the geological map defines the strike of the unconformity (Figure 7a) and the fault orientation has been measured (the normal to the fault $\boldsymbol{n}_f$ is available). However, in this fictitious scenario, the dip of the unconformity plane (hence its normal $\boldsymbol{n}_u$) is not known and needs to be recovered. We assume that the erosion is planar, so that there is only a lack of knowledge for dip of the unconformity (equivalently, the vertical component of $\boldsymbol{n}_u$). The objective of this simple synthetic test is thus to estimate how accurately the unconformity can be modelled, and consequently, to determine whether we can retrieve the vertical component of $\boldsymbol{n}_u$ and the model that gave rise to the observed measurements.

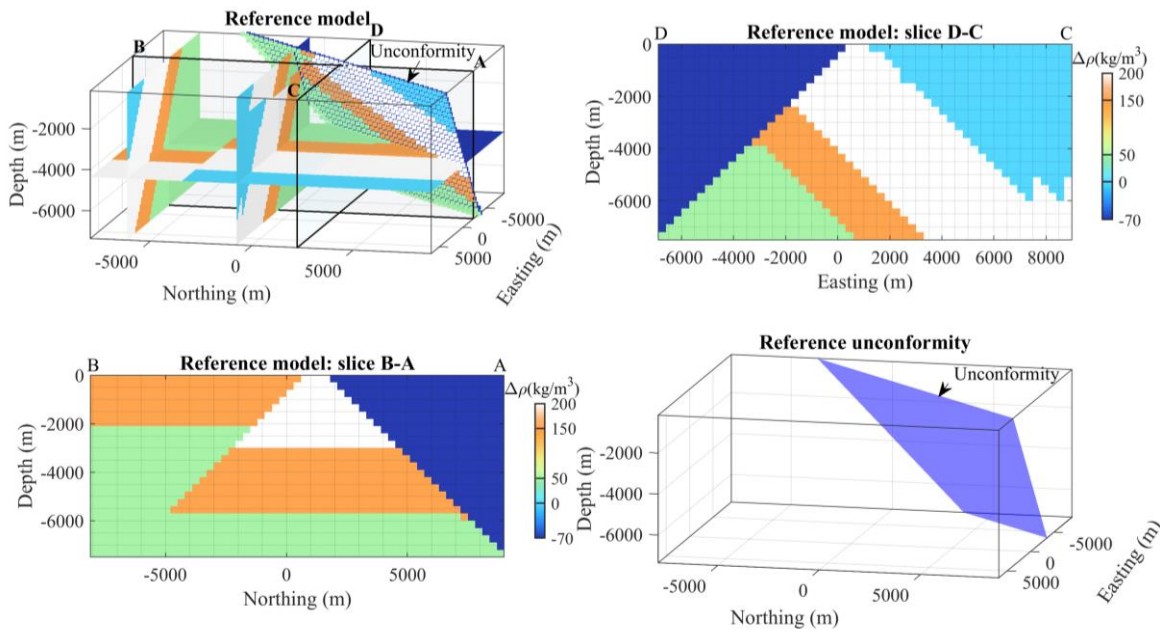

Figure 8. Reference model visualised in 3D view, slices, and representation of the unconformity plane.

To test the geological components of inversion introduced in this paper, four geophysical inversion scenarios are investigated:

(1) No use of implicit geological modelling during inversion.

(2) Geological modelling used to calculate a geological correction term to model updates (Sect. 3.3.1).

(3) Geological modelling used only to define the prior model term in the cost function of geophysical inversion (Sect. 3.3.2).

(4) Geological modelling used in both the definition of the prior model term in the cost function and to calculate a geological correction term to model updates.

For the recovery of the unconformity plane (erosional surface), we separate the rock units into pre-erosional and post-erosional stratigraphic groups. The location of contacts between these two groups is used for the calculation of a plane defining the unconformity as detailed in Section 3.2.2. Assuming a complete lack of knowledge about the vertical component of $\boldsymbol{n}_u$, we set it to 0 (i.e., vertical unconformable contact) in the starting model for inversion (Figure 9). We run inversion corresponding to the four inversion scenarios proposed above. Inversions stop when reaching $ERR_d = 0.5$ mGal, which corresponds to acceptable values for legacy data (Barnes et al., 2011). The gravity data simulated for the reference and starting models are shown in Figure 10. We consider the data produced by the reference model as the field measurements corresponding to the model we try to recover. We assume zero error to test the ability of the method to recover the reference in a perfect data settings (See Appendix 3: Robustness to noise and Appendix 4: Robustness to a degenerate starting model for a more realistic case including data errors).

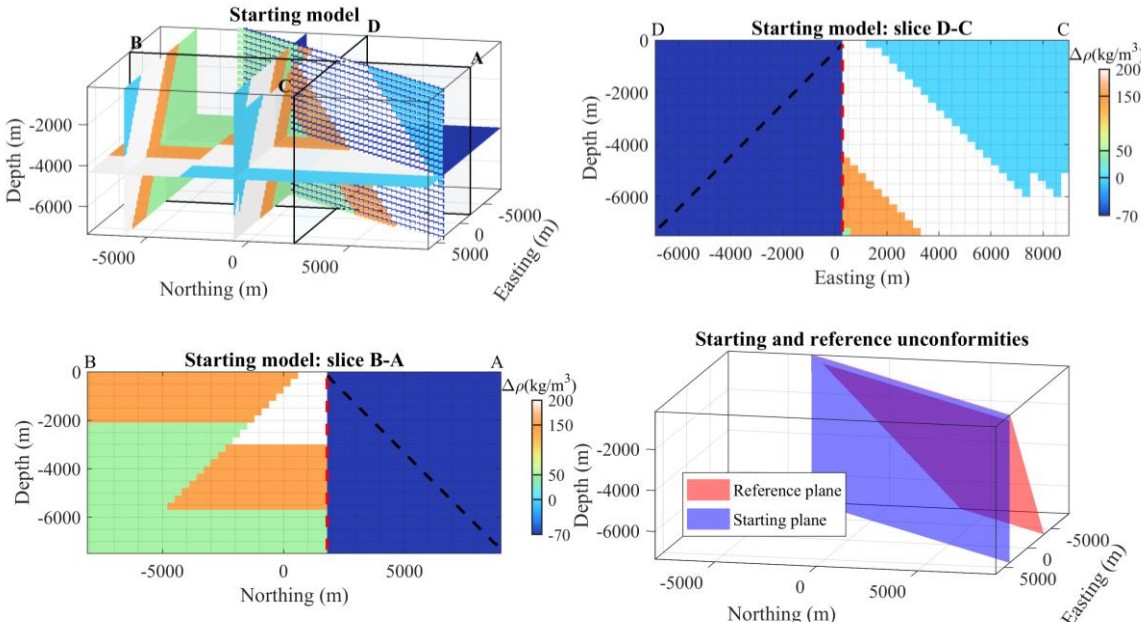

**Figure 9. Starting model visualised in 3D view, slices, and representation of the unconformity plane. The black dashed line represents the location of reference (or true) unconformity plane.**

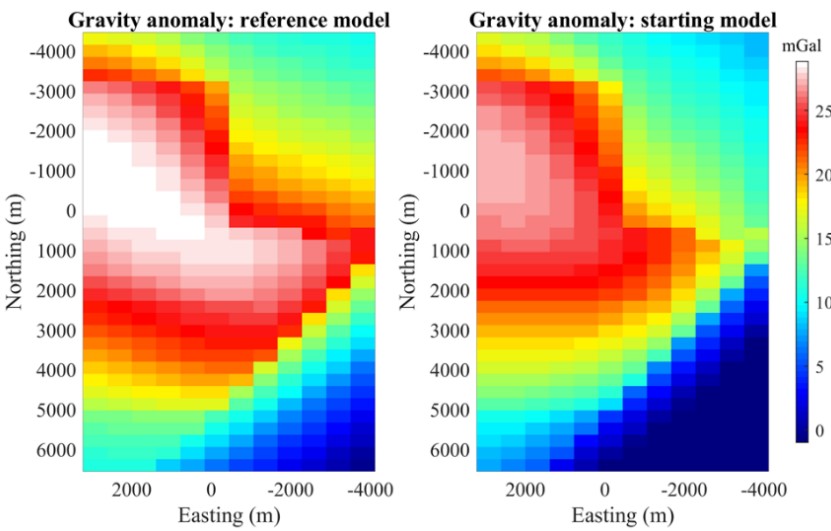

**Figure 10. Gravity data simulated for the reference model (left), which we consider as the field data to honour, and for the starting model (right), which corresponds to the starting point of all inversion scenarios.**

### 5.2.2. Inversion results

As a pre-requisite to comparing models, we point out that they are all geophysically equivalent from the point of view of the data misfit $ERR_d$ (Figure 13a). On this basis, the features presented by the recovered models can be assessed from a

505 geological and petrophysical perspective. Starting with visual, qualitative interpretation of the results of the four scenarios listed above, we examine:

- The 3D model in perspective with the unconformity shown as scattered points in Figure 11;
- The slices B-A and D-C in Figure 12.

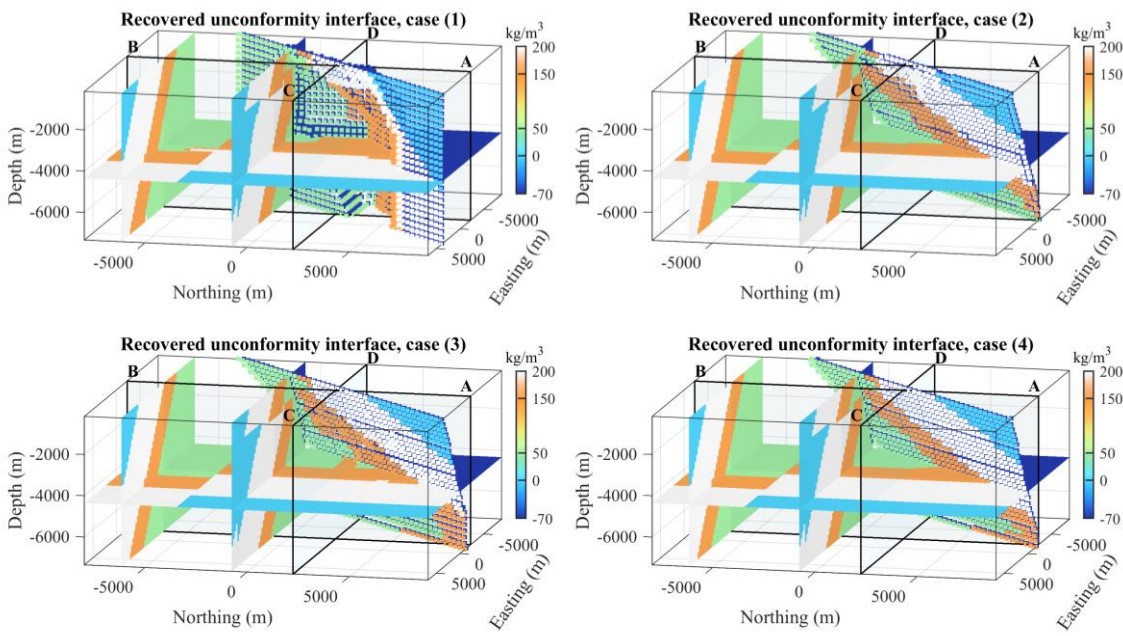

**Figure 11. 3D visualisation of Inversion results for cases (1) through (4). The differences between the best fitting planes corresponding to the recovered uniformity interface shown here and the reference plane (Fig. 9) amount to 11.9, 3.2, 5.7, and 4.6 degrees for cases (1) through (4), respectively.**

The main observation that can be made is that the unconformity boundary is poorly recovered for case (1) in the absence of either geological correction or a geological prior model term applied to inversion. This is clearly visible in all images showing inversion results, be it from the 3D plot of points constituting the non-conformable contact (Figure 11) or slices through the model (Figure 12). This observation is further confirmed by the metrics shown in Figure 13. Case (1), which considers geophysical data only in the inversion process, stands out for all metrics. For instance, $ERR_m$ remains notably higher when no geological modelling is used in the inversion than for all other cases (Figure 13c). A similar behaviour is observed both for $OC$ and $ERR_\Phi$ (Figure 13b and Figure 13d, respectively).

From this preliminary examination of results on a relatively simple geological case, we conclude that using automatic geological modelling in inversion can dramatically increase the inversion's capability, not only to recover models consistent with geological data and principles, but also to avoid converging to local minima when the starting model is inappropriate. This is true for both the application of a geological correction term or for the definition of a dynamic prior model term. Further to this, convergence curves in Figure 13a suggest that inversion considering geophysics alone may require many more iterations to converge as compared to using appropriate geological correction.

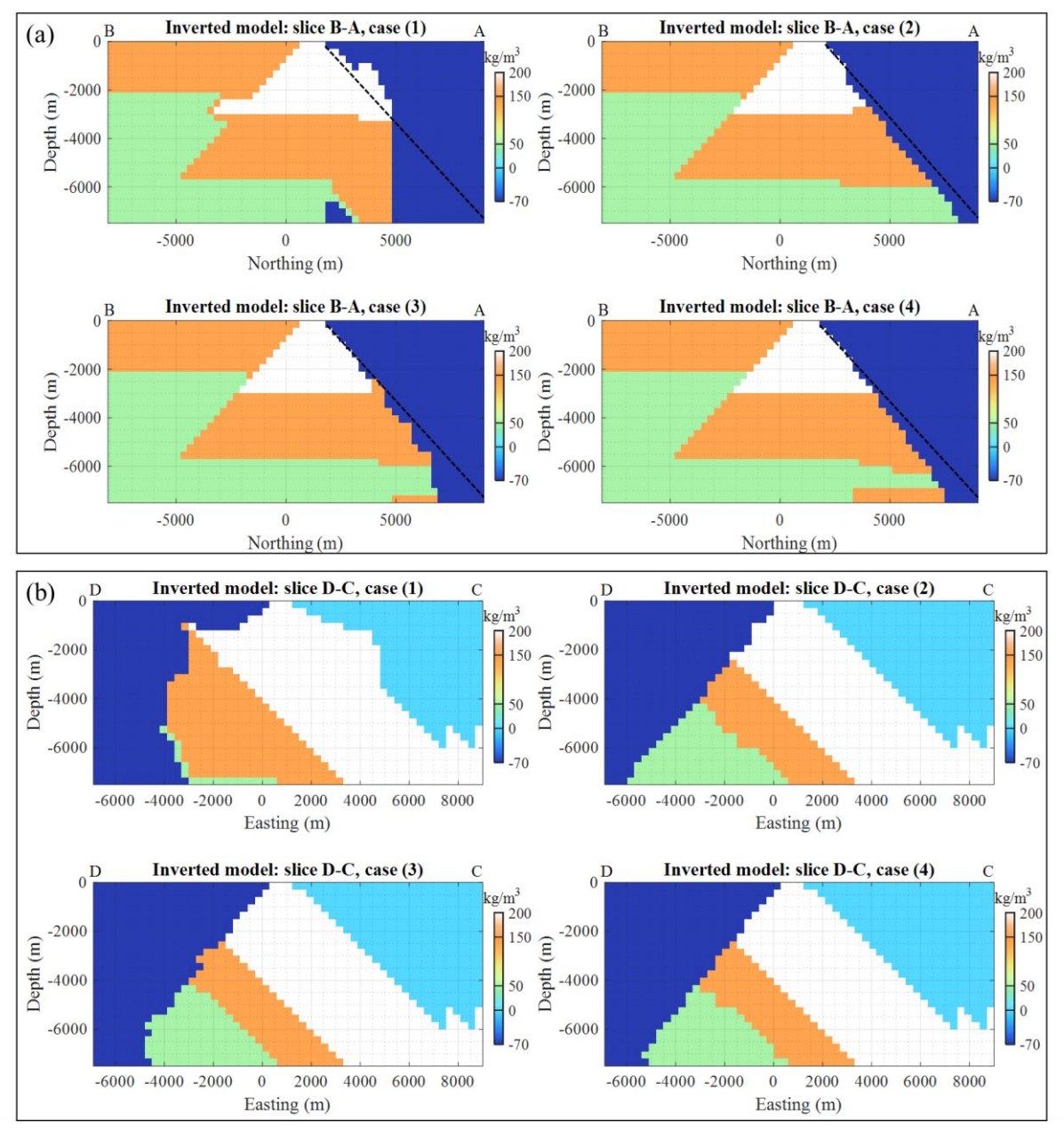

**Figure 12. Inversion results for cases (1) through (4) along section B-A (a) and D-C (b). In (a)The true location of the unconformity is indicated by the black dashed line.**

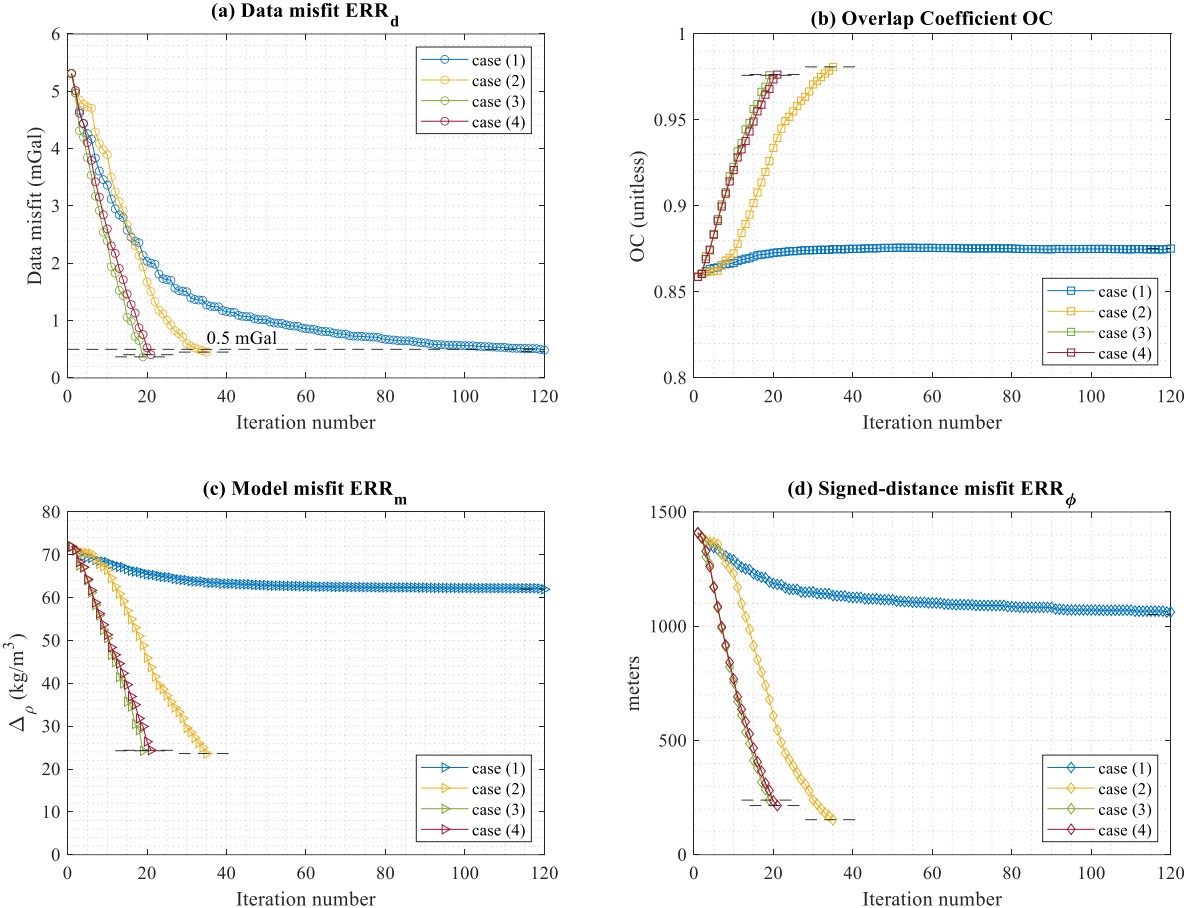

**Figure 13. Inversion metrics for cases (1) through (4): data misfit (a), Overlap Coefficient (b), model misfit (c) and signed-distance misfit (d).**

Visually comparing models from cases 2, 3 and 4, we notice features largely similar to the reference model, but also some fine scale differences. Scenario 2 presents a contact between the non-conformable unit and the sequence that is slightly better recovered at depth than for scenarios 3 and 4 (Figure 12). Cases 2, 3, and 4 are difficult to distinguish visually except in the deeper part of the model, but this may be inconclusive due to the limited geological and geophysical sensitivity in this part of the model. Overall, visual inspection suggests that scenario 2 seemingly has a higher degree of resemblance with the reference model while converging to a similar geophysical misfit. This is also suggested by the calculation of the dip angle of the recovered unconformity plane by automatic interpretation of the best fitting plane. The difference with the reference model amounts to approximately 11.9, 3.2, 5.7, and 4.6 degrees for cases (1) through (4). Taken together, our results using this example suggest that:

- The use of geological modelling to define either a dynamic prior model or a correction term greatly increases the geological realism of the final model;
- Inversion using a geological prior model term converges faster than otherwise;
- The use of a geological correction term may provide slightly better recovered unconformity planes.

### 5.3. Improving the geological realism of a pre-existing model

In this section, we investigate the possibility of increasing the geological realism of a pre-existing model provided a priori from, e.g., already performed inversion or classification of inversion results. To simulate this, we first start from the model inverted with the level set method without geological correction as obtained in Section 5.1 (Figure 14b), and run inversion

with geological correction applied. The inverted model obtained in this fashion is shown in Figure 14a. An animation showing the evolution of the inverted model together with geological inconsistences is shown in supplementary material provided by Giraud and Caumon (2023). The application of geological correction manages to remove a number of unrealistic features present in the starting model. However, the effect of the geological correction seems visually smaller than in the application of geological correction from the onset of inversion (Figure 14c). On this premise, if large unrealistic features appear, we recommend to run geological correction from the beginning of inversion instead of as an ad-hoc process. We note that in the transition between Figure 14a and Figure 14b by application of geological correction, all models are geophysically equivalent or nearly equivalent in that they present similar geophysical data misfit values. This suggests that, for this dataset, a continuum of models exists fitting the geophysical data to a similar level while presenting different degrees of geological realism. In what follows, we investigate this possibility further using the synthetic dataset presented in Sect. 5.2

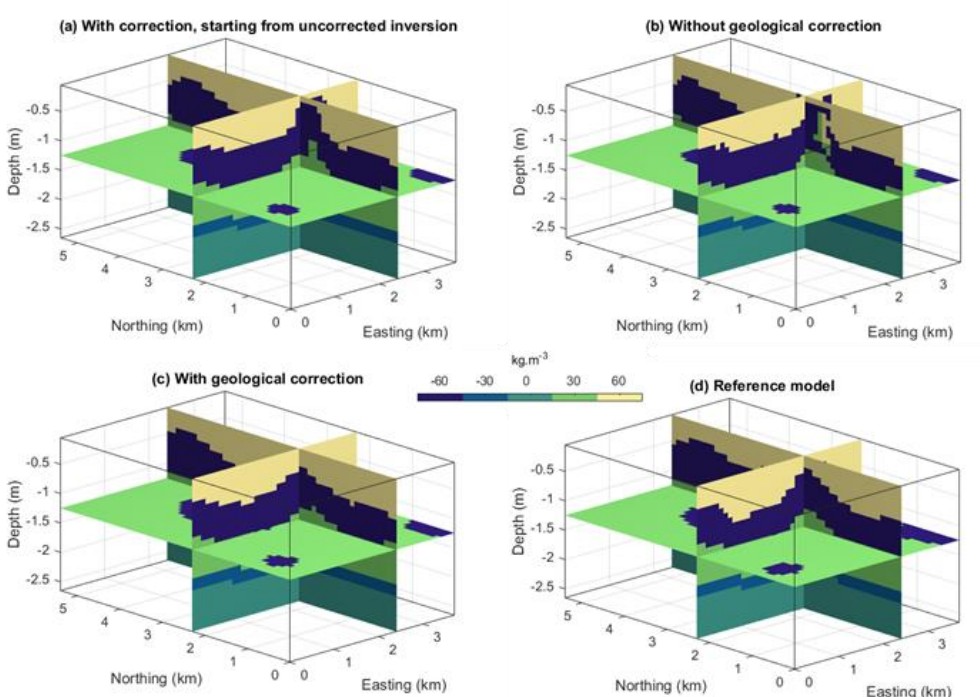

**Figure 14. Comparison of inversion results with (a) geological correction starting from inverted model obtained without geological correction, as shown in (b); results with geological correction (c) and reference model (d).**

The starting model we consider for improving the geological realism (Figure 15a) shows a very convoluted geometry for the unconformity and significant thickness variations in the pre-erosional sequence, while presenting a geophysical data misfit value $ERR_d$ close to the objective data misfit value of 0.5 mGal (dashed line in Figure 13a). In real world studies, this could correspond to the results of rock type classification obtained a posteriori from geophysical inversion only or legacy inversion. We will further refer to this case scenario as case 5. As can be seen in Figure 15a, the starting model corresponding to case 5 is in strong disagreement with the reference model (Figure 8) from a structural geological point of view as shown, for instance, by the high starting $OC$ value (Figure 16b). In particular, the unconformity is poorly recovered and the expected planar contact is significantly distorted. In contrast, the layered stratigraphy is well resolved.

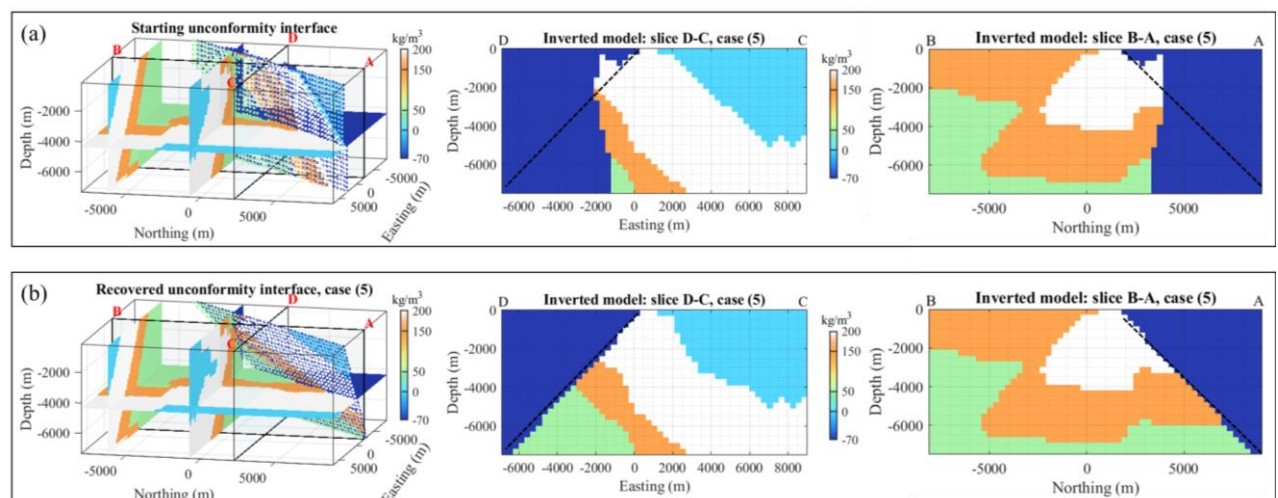

**Figure 15. (a) starting model and (b) inverted model. From left to right, the views are the same as displayed in Figure 11, and Figure 12, respectively.**

We use this model as a starting model for inversion applying geological correction only (Section 3.3.1, Eq. 4 and flow ① in Figure 2). We use this scenario to evaluate the capability of our method to restore geological consistency between inversion results and geological observation while maintaining geophysical data fit within prescribed levels. As in the previous case study, we focus the analysis on the unconformity since it is the main feature targeted by the inversion in this example.

Visual inspection of inverted models shown in Figure 16b indicates that the application of geological correction effectively drives the optimization process towards models in better agreement with the unconformity observed in the geological map. However, the thickness variations in the pre-erosional sequence are only partly resolved. Nonetheless, the resulting model is in a region of the model space considerably closer to geologically plausible scenarios than the starting model. This is illustrated by the metrics used to monitor the inversion, which show an overall increase in $OC$ and a decrease in model misfit (Figure 16b and Figure 16c, respectively). This implies that using a geological correction term may reduce the risk of geophysical inversion converging to geologically unrealistic local minima of the cost function.

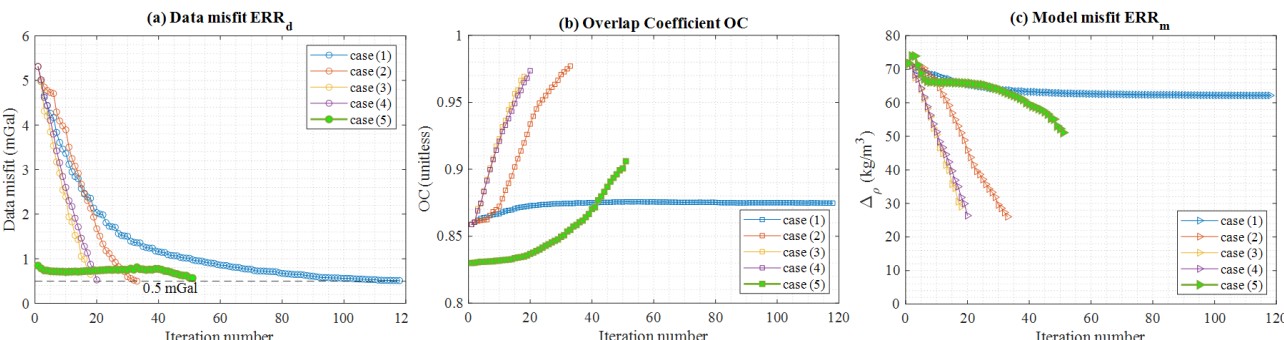

**Figure 16. Comparison of inversion metrics for cases 1-4 with case 5.**

In terms of geophysical data misfit, the data error $ERR_d$ presents a plateau decreasing only after iteration 40 (Figure 16a), indicating gradual deformation of models with similar $ERR_d$ values. This shows that inversion navigates a region of the model space comprising models that are equivalent in terms of geophysical data misfit but which are gradually more consistent with the available geological data. This implies that the proposed approach may be effectively used to navigate the space of geophysically equivalent models. We note that the exploration of geophysically equivalent models can be performed using null-space shuttles to modify an already existing model while maintaining a nearly constant geophysical

data misfit (Deal and Nolet, 1996; Muñoz and Rath, 2006; Fichtner and Zunino, 2019). In our case, it could be achieved by gradual deformation of the starting model to accommodate geological information. On these premises, the use of geological correction as performed in this section may opens up avenues to:

- Derive models not proposed by implicit geological modelling or geophysical inversion alone but combining them instead.
- Infuse geological data and principles into pre-existing inverted models derived without the knowledge or capability to integrate them.

## 6. Discussion

### 6.1. Proposed work in the context of geoscientific exploration

In the last decade, the sampling of models from regions of the geological solution space to fit geological field observations has gained traction. The models proposed by these methods are usually derived from the interpolation of geological observations (Wellmann and Caumon, 2018). The inversion algorithm we proposed here can be used in conjunction with this strategy to modify such models using geophysical inversion. As geophysical inversion may be sensitive to features geological modelling has little to no sensitivity to, geophysical inversion can be used to explore different regions of the solution space. This may be achieved with little development using models representative of families sampled from the geological model space, e.g., from topological analysis (Pakyuz-Charrier et al., 2019) or a similarity distance (Suzuki et al., 2008) to select starting models for inversion. Following the same idea, it may be possible to use deep learning for 3D geological structure inversion results (Jessell et al., 2022; Guo et al., 2021) as starting points to run series of inversions using the method presented here.

In some cases, geological modelling and geophysical inversion are difficult to reconcile. This may indicate that the modelling of geology is not sufficiently well informed by the available geological measurements and/or that hypotheses about the area need to be revisited. Under these circumstances, the geological model space might not contain a sufficiently good representation of the subsurface to fit the geophysical data. It could be the case, for instance, when structures are invisible to the available geological observations but can be sensed by geophysical data. In such cases, geophysical inversion as performed here could be used as a tool to adjust these models and to infer the presence of unseen geological features.

As mentioned above, our inversion algorithm enables estimating the magnitude of adjustments required to reconcile geological models with geophysical measurements not only by comparing the forward response of given models but also, more importantly, by adjusting the model's geometry to fit geophysical data. For scenario testing, the parameter controlling the amplitude of perturbation of interfaces between two successive iterations, $\tau$, can be set accordingly with uncertainty information. For instance, it can be set arbitrarily small in locations of low uncertainty such as the vicinity of boreholes, outcrops, or high resolution seismic. Following the same idea, $\tau$ could also be set using uncertainty about domain boundaries derived using the implicit approach of Fouedjio et al. (2021). Further considerations of uncertainty may be required to better evaluate and understand inversion results. For instance, the approach of Wei and Sun, 2022, who generate series of inverted models by varying their deterministic inversions' hyperparameters could be a source of inspiration for uncertainty estimation. Likewise, the scalar field perturbation of Henrion et al. (2010), Clausolles et al., (2023) or Yang et al. (2019) could be transposed to the modelling approach we propose here.

We note that while experienced interpreters may be able to interpret geologically unrealistic inversion results 'correctly', it is nonetheless safer to ensure geological principles are not violated by inversion. It evacuates an important source of uncertainty, and it reduces the impact of human bias and ambiguous interpretation while ensuring that results can be robustly passed on to the next stage of modelling. More fundamentally, adding constraints essentially reduces the non-uniqueness of the inverse problem by making the search space smaller. In practice, however, more studies should be performed to assess whether the geological constraints change the "landscape", or rugosity, or the objective function, and whether it always helps optimization methods to avoid convergence to local minima.

Finally, the approach presented here belongs to a family of inversion approaches that could be referred to as 'geometry driven'. That is, the main driver for fitting the geophysical data is the geometry of the subsurface model. One of our working assumptions is that geological data used to derive structural geological models are complemented by geophysical data, with the possibility to alter the shape of geological models. Another 'geometry driven' approach, consists in sampling points controlling the interpolated geological models within uncertainty and to integrate their forward geophysical response to the calculation of a posterior distribution (Güdük et al., 2021, Liang et al., 2023). This method is very elegant as it uses a unique geological level set parameterization, but it does not explicitly address the integration of spatial geological data as provided by, e.g., drillholes. In contrast, our method can honour (up to discretization errors) spatial data. The mapping of the geophysical signed distances $\phi$ onto their geological counterparts through $f^{geol}$ makes it challenging to derive sensitivities directly as in Liang et al (2023), but it probably makes it possible to explore a larger model space by possibly departing from geological acceptable solutions during the non-linear iterations. Although further tests and comparisons between the two approaches are probably needed, this feature could be useful to prevent the optimization from converging to local minima and could possibly be used in the future for stochastic inversions.

## 6.2. Limitations of the method and potential weaknesses

A potential limitation of the proposed approach that is inherent to the use of geological constraints pertains to the projection of an implicit rock model realisation onto the discrete mesh used to model geophysical data. Quoting Scalzo et al. (2021): "Naively exporting voxelised geology […] can easily produce a poor approximation to the true geophysical [response]". While it can be alleviated with the appropriate material averaging approach, aliasing can affect the geophysical response of the geological model projected onto the aforementioned mesh. This consideration is absent from our investigations, but it may be important to address in real world case studies where stakes are higher than in the work presented here.

A clear limitation of this work, which is intrinsic to level-set inversion, is the discreteness of physical properties inverted for, which probably makes this kind of inversion one of the most parsimonious approaches. It presupposes accurate knowledge of the density of rock units and/or that they can be described by constant densities, neglecting geological and petrophysical phenomena leading, for instance, to lateral facies variations, compaction, and alteration. It is therefore possible for several rock types from the same stratigraphy to fall under the same 'unit' in the modelling approach that we follow. Polynomials describing properties could be used instead of constant values. This limitation precludes application of the method in certain scenarios. This may also call for integration with more advanced statistical rock physics and geostatistical modelling, using, for instance, the approach of Phelps (2016) to generate densities using a geostatistical approach. Additionally, porting the continuous-value inversion component of the shape optimization workflow of Dahlke et al. (2020) to 3D potential field inversion may help to generalise the potential use of level set inversion. Alternatively,

the mapping from signed distances to a fixed density value used here could be replaced by a mapping to an interval defining bounds constraints enforced during inversion using the same approach as, e.g., Ogarko et al. (2021).

Finally, as mentioned in several places in the paper, further considerations on uncertainty are also required to better understand inversion results and find a set of admissible models, both geologically and geophysically.

### 6.3. Future research avenues

We formulate and solve the inverse problem in the least-squares sense using Tikhonov regularisation to stabilise the inversion and to infuse geological information. The regularisation functional we employ addresses only the difference between the proposed model and a given model. It is straightforward to consider other regularisation terms such as a gradient or Laplacian minimization to encode geological information. Another avenue could be to use the same parametrisation for the geophysical and geological modelling, and to solve for all information simultaneously.

In addition, the flexibility offered by the least-squares framework allows for the design of constraint terms specific to the use of signed distances to enforce physical principles. For instance, one can think of maintaining the sum of the updates of signed distances to zero in each model-cell, such that, for the $i^{th}$ model-cell, $\sum_{k=1}^{N} \delta\phi_k^i = 0$. This would preserve the signed-distance properties of $\phi$ at each iteration, thereby eliminating the need to reinitialise the signed-distance fields at each iteration using the fast-marching method.

The available geological information to constrain geophysical inversion can take many forms ranging from sparse information about rocks properties to dense borehole data and seismic interpretations (Grana et al., 2012). We have explored the possibility to constrain a geophysical inverse model using surface geological data and seismic sections. Borehole information can, without doubt be considered in the same manner, and that there is no restriction to the use of other sources of information such as modelling of other geophysical techniques (e.g., depth-to-basement from electromagnetic methods, reflectors from passive seismics, etc).

An obvious and straightforward extension of this work is to extend it to magnetic data, the inversion of which shares many features with gravity inversion.

Finally, extensions of our method may allow for null-space exploration and the mitigation of some of the limitations identified in the previous subsection. This paper investigates the importance of geological information in level set inversion. Previous work focusing on level set inversion following an approach similar to ours have investigated the importance of accurate knowledge on the geometry and the number of rock units a priori (Giraud et al., 2021a). Giraud et al., 2021a, and Rashidifard et al., 2021, suggest that inversion is somewhat robust to errors in the starting model geometry and in the petrophysics of the rock units. Nonetheless, relatively small deviations between scenarios 2 and 5 illustrate that the proposed methodology is not sufficient to address the ill-posedness of the potential field problem. Moreover, results from Giraud et al., 2021a, suggest that level-set inversion "presents limitations when an important geologic unit is missing from the initial model". To alleviate this, ways to generate the 'birth' of new geological units for inversion to consider geological bodies previously not accounted due to lack of information may be devised. One possibility could be to use the sensitivity of geophysical data to changes in physical property. This may be useful, for instance, to model intrusions invisible to surface geology. Further to this, starting from a discrete model resulting from level-set inversion, it is possible to explore regions of the model space with intra rock unit variations (lateral facies variations, compaction, etc.) simply through usage of the null-space shuttles as proposed by Deal and Nolet (1996) and Fichtner and Zunino (2019). We believe that these two possibilities are sides of the same coin that constitute a promising area of research for future work on uncertainty quantification.

## 7. Conclusions

We have introduced two novel approaches towards unification of geological modelling and geophysical inversion into a deterministic inversion algorithm. They consist of:

- The application of a geological correction term to the geophysical inversion's model update, and
- The integration of automated geological modelling in a dedicated term of the objective function governing geophysical inversion,

and allow to obtain inverse models that are consistent with geological principles and data.

These developments were motivated by the need to remediate some of the limitations inherent to geological and geophysical modelling taken separately. We have shown that our framework is general in nature and can be applied in different contexts. We have tested the proposed approaches and demonstrated their potential using two synthetic examples, each representing a specific exploration scenario:

- to constrain the modelling of a conformable stratigraphy, which relates to the deposition order of sediments,
- to recover the parameters of a tilted unconformity, which relates to tectonic history.

In both cases, geological information used to derive constraints is sparse, and separate geological or geophysical modelling do not suffice to recover the reference model. We have shown experimentally that, in contrast, the integration of the two sources of information as proposed here provides the capability to recover models much closer to the reference structures and reduces the effect of the ill-posedness affecting each method. Our investigations also suggest that the methodology we propose can be applied with different goals in mind, including:

- Complementing geological sampling techniques by automatically tuning implicit geological models to fit geophysical data.
- Deriving models as starting points to navigate the joint geology-geophysics null-space.
- 'Geologifying' pre-existing models and exploring the geophysical space of equivalent models.
- Recovering geological parameters and models in sparse data scenarios.

## 8. Data and code availability

LoopStructural was made publicly available by Grose et al. (2020); the latest version is available at: https://github.com/Loop3D/LoopStructural (last accessed on 30/07/2023). The geological data used to generate models and models shown here are provided in Giraud et al. (2023). The inversion used here is a prototype under development that will be released in the future. Supplementary material is made available by Giraud (2023) and Giraud and Caumon (2023).

## 9. Authors contribution

JG: conceptualisation with contributions from GC and LG, modification of original geological models generated by LG, geophysical modelling and inversion, with peer review of results with GC, development and testing of the geophysical inversion code, development and testing of the link between geophysical inversion and geological modelling, result analysis, redaction of manuscript with comments from all other authors, funding acquisition.

LG created the geological model then modified by JG, reviewed the manuscript and contributed to the conceptual design of the method. PC took part to discussions with JG and GC, especially on model space exploration, and he contributed to the redaction of the paper; VO provided critical review of the manuscript and extensive comments and suggestions.

## 10. Competing interests

The authors declare that they have no competing interests.

## 11. Acknowledgements

Jérémie Giraud acknowledges support from the European Union's Horizon 2020 research and innovation programme under the Marie Skłodowska-Curie grant agreement No. 101032994. The authors also thank the RING (Research for Integrative Numerical Geology) and Loop Consortia for their support. Vitaliy Ogarko acknowledges support from the Mineral Exploration Cooperative Research Centre, whose activities are funded by the Australian Government's Cooperative Research Centre Program. This is MinEx CRC document 2022/83.

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

## 13. Appendices

### 13.1. Appendix 1: Building the least-squares system of equations

We derive the system of equations solved in the level-set inversion scheme used here. Writing $\boldsymbol{d}^{calc} = \boldsymbol{S}^m \boldsymbol{m}(\boldsymbol{\phi})$, the system of equations corresponding to Eq. (4) is given as:

$$\begin{bmatrix} \boldsymbol{S}^m \boldsymbol{m}(\boldsymbol{\phi}) \\ \lambda_p^2 \boldsymbol{W}_p \boldsymbol{\phi} \end{bmatrix} = \begin{bmatrix} \boldsymbol{d}^{obs} \\ \lambda_p^2 \boldsymbol{W}_p \boldsymbol{\phi}^{prior} \end{bmatrix}, \tag{18}$$

At each iteration, the system is linearised around the current model. Solving for updates of the signed distances $\delta\boldsymbol{\phi}$, we obtain, for the $k^{\text{th}}$ iteration:

$$\begin{bmatrix} \boldsymbol{S}^m \dfrac{\partial \boldsymbol{m}(\boldsymbol{\phi}^k)}{\partial \boldsymbol{\phi}^k} \\ \lambda_p^2 \boldsymbol{W}_p \end{bmatrix} \delta\boldsymbol{\phi} = - \begin{bmatrix} \boldsymbol{S}^m \boldsymbol{m}(\boldsymbol{\phi}^k) - \boldsymbol{d}^{obs} \\ \lambda_p^2 \boldsymbol{W}_p (\boldsymbol{\phi}^k - \boldsymbol{\phi}^{prior}) \end{bmatrix}. \tag{19}$$

Using the chain rule as in Eq. (5), we can rewrite the first element of the left hand side of Eq. (23) as follows and obtain the sensitivity matrix of the gravity data to changes of $\boldsymbol{\phi}$:

$$\boldsymbol{S}^\phi = \boldsymbol{S}^m \frac{\partial \boldsymbol{m}(\boldsymbol{\phi}^k)}{\partial \boldsymbol{\phi}^k} = \frac{\partial \boldsymbol{d}}{\partial \boldsymbol{m}(\boldsymbol{\phi}^k)} \frac{\partial \boldsymbol{m}(\boldsymbol{\phi}^k)}{\partial \boldsymbol{\phi}^k}. \tag{20}$$

$\boldsymbol{S}^\phi$, which relates the perturbation of signed distances to geophysical data, can be calculated from Eq. (2-3) (see Giraud 925 et al., 2021a, for details about this derivation). Using Eq. (20), the system of equations in Eqs. (18-19) rewrites as:

$$\begin{bmatrix} \boldsymbol{S}^\phi \\ \lambda_p^2 \boldsymbol{W}_p \end{bmatrix} \delta\boldsymbol{\phi} = - \begin{bmatrix} \boldsymbol{S}^m \boldsymbol{m}(\boldsymbol{\phi}^k) - \boldsymbol{d}^{obs} \\ \boldsymbol{W}_p (\boldsymbol{\phi}^k - \boldsymbol{\phi}^{prior}) \end{bmatrix}. \tag{21}$$

$\boldsymbol{W}_p$ is given as:

$$W_p = \begin{bmatrix} W_p^{\phi_1} & 0 & 0 & 0 \\ 0 & W_p^{\phi_2} & 0 & 0 \\ 0 & 0 & ... & 0 \\ 0 & 0 & 0 & W_p^{\phi_{n_r}} \end{bmatrix}, \tag{22}$$

where $W_p^{\phi_k}$ is a diagonal matrix of dimensions $n_m \times n_m$ locally adjusting the amount of change allowed to the signed-distance updates of the different model-cells.

It is calculated as:

$$W_p^{\phi_k} = \Sigma_p \cdot (\mathbb{1}_\tau(\boldsymbol{\phi}^k))^T, \tag{23}$$

where $\Sigma_p$ is a diagonal matrix containing uncertainty information derived from prior information. It can be used to control where the model updates are prioritised over other locations. In the absence of such prior information, $\Sigma_p$ may be set as the identity matrix; $\mathbb{1}_\tau(\boldsymbol{\phi}_k)$ is the indicator vector of $\boldsymbol{\tau}_i$ applied to $\boldsymbol{\phi}_k$. For the $i^{\text{th}}$ model cell, it is calculated as:

$$(\mathbb{1}_\tau(\boldsymbol{\phi}_k))_i = \begin{cases} 1 \ if \ |(\boldsymbol{\phi}^k)_i| \le \boldsymbol{\tau}_i \\ 0 \ if |(\boldsymbol{\phi}^k)_i| > \boldsymbol{\tau}_i \end{cases} \tag{24}$$

As a consequence, $W_p^{\phi_k}$ is equal to 0 for all cells not part of the inversion's domain (i.e., cells where $\mathbb{1}_\tau(\boldsymbol{\phi}_k)$ is null) and

to 1 for all cells considered at any given iteration.

### 13.2. Appendix 2: Estimating the best fitting plane of a contact under constraints

In this Appendix, we detail the calculation of the vector determining the orientation of an erosion plane as mentioned in section 3.2.2. It is calculated from the location of the contacts between rock units making up the modelled unconformity. Let us define a plane by its normal vector $\boldsymbol{n} = [n_x, n_y, n_z]$ and a real number $s$ such that $\boldsymbol{nx} + s = 0$, where $\boldsymbol{x}$ is a

coordinate vector in three-dimensional space. The best fitting plane approximating the contact between two units or groups of units is obtained by solving a linear equality-constrained least squares problem formalised as follows:

$$\text{minimise } \|\boldsymbol{WAn} - \boldsymbol{Wb}\|_2^2 \tag{25}$$
$$\text{s.t. } \boldsymbol{Cn} = \boldsymbol{d},$$

where $\boldsymbol{b}$ is a column vector of ones and $\boldsymbol{W}$ is a diagonal matrix. It contains weights controlling the relative importance given to the different values in $\boldsymbol{A}$, which contains the spatial coordinates of the points constituting the interface to be approximated as a planar contact. The matrix $\boldsymbol{C}$ contains the location of contacts measured, e.g., at surface level from

geological observations or from borehole data, which are used to define the equality constraints; $\boldsymbol{d}$ is a vector of ones. In our implementation, we solve Eq. (25) using the open-source linear algebra library LAPACK proposed by Anderson et al. (1999), to which we refer the reader for further details.

In practice, $\boldsymbol{W}$ can be set according to the sensitivity of geophysical data to variations in physical property (sensitivity matrix $\boldsymbol{S}$) in the model-cells considered. To constrain the horizontal component of the normal vector calculated using the

system of equations in Eq. (25), 2 points located at surface level suffice. In this situation, we obtain an estimate of the third component of the normal vector to the plane, $n_z$.

After obtaining the normal vector approximating the orientation of the plane, $s$ can be determined from a location that the plane is known to cross (i.e., at the modelled interface) using the definition of the plane as $\boldsymbol{nx} + s = 0$.

### 13.3. Appendix 3: Robustness to noise

We investigate the robustness of inversion with geological correction to random Gaussian noise, using the synthetic dataset introduced in Sect. 5.1.

We add noise with zero mean and a standard deviation equal to 0.075 mGal. This corresponds to 8% of the absolute difference between the lowest and highest gravity anomaly value in the simulated dataset. We note that this value is superior to the case for a 'carefully acquired and corrected' land survey, where a value of 0.05 mGal is acceptable (Barnes

et al., 2011). The dataset with noise contamination, the forward data corresponding to the inverted model, their difference (i.e., the misfit map), and the noise that was added to the data are shown Figure 17.

While the patterns that are visible in the difference map seem to be largely random, there might be a non-random component to the difference map possibly due to the incomplete deformation of geological interfaces. We do not reproduce the resulting model as it is visually largely similar to the case without noise in Figure 5b.

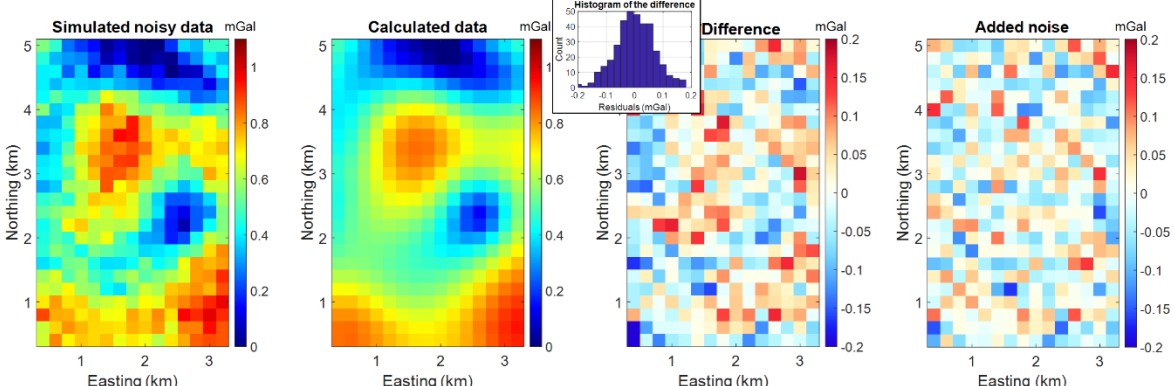

**Figure 17. Simulation of noisy data. From left to right: data contaminated by noise, calculated data from the inverted model, difference between inverted and calculated data with histogram and noise that was added to the data..**

### 13.4. Appendix 4: Robustness to a degenerate starting model with noisy data

In this appendix we are interested in testing the robustness of the method to changes in the starting model. For this, we use the same noisy dataset as in Appendix 3: Robustness to noise and simulate a degenerate starting model. For this, we rotate the starting model by 180 degrees around the vertical axis. The so-obtained starting model and inverted model are shown in Figure 18a and Figure 18b, respectively. The starting data corresponding to this starting model is shown in Figure 18c. When compared with the simulated field data (Figure 18d), it is clear that this starting model is degenerate

and present an extreme scenario. Nevertheless, inversion converges to a stable solution and manages to recover some of the features of the true model. However, the difference between the data inverted for (Figure 18d) and the calculated data (Figure 18d) shows non-random patterns (Figure 18f). This indicates that the inversion might be stuck in a local minimum and is unable to fit the data appropriately. Notwithstanding, this confirms that our approach is robust to errors in the starting model and to the presence of noise in the data. This also shows that in this example, the interaction between

geological modelling and geophysical level set inversion leads to a reasonable solution with some realistic features, even though it is not sufficient to completely disambiguate the geophysical inverse problem.

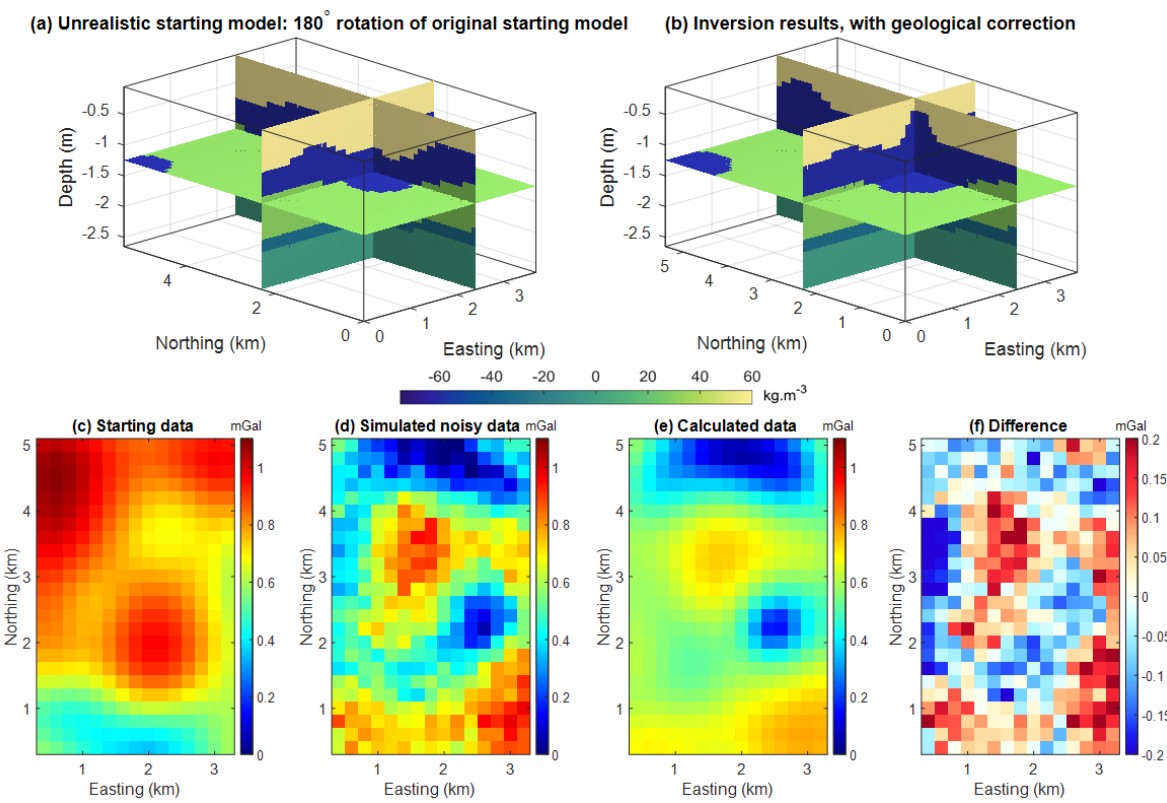

**Figure 18. Simulation of a degenerate case. Top row: starting (a) and inverted model (b). Bottom row: starting data (c), data contaminated by noise (d), calculated data from the inverted model (e), and difference between inverted and calculated data (f).**

### 13.5. Appendix 5: Robustness to errors in the density of rock units

In this appendix, we investigate the impact of inaccurate estimations of the density of rock units using the synthetic model presented in Sect. 5.1. We generate two starting models, considering the density contrast of unit 4 (dark blue unit in Figure 4 and all Figures showing this model):

- In the first starting model, the absolute density contrast value of is overestimated by 15 kg/m$^3$.
- In the second starting model, the absolute density contrast is underestimated by 15 kg/m$^3$.

This leads to using density contrasts of -75 kg/m$^3$ and -45 kg/m$^3$, respectively, instead of -60 kg/m$^3$. In this example, we use gravity data without noise contamination. Results are shown in Figure 19, where we also remind, for comparison, the results obtained for the other tests we performed using this model. Figure 19b shows the inverted model using -75 kg/m$^3$ for unit 4, which leads to a reduced overall volume of rock for unit 4. Figure 19b shows the inverted model with -45 kg/m$^3$, which leads to an increased overall volume of rock for unit 4. In both cases, the overall geometry of the rock units are preserved when compared to the other cases and the reference model.

We note that this observation is in line with Giraud et al., 2021a, who do not consider the case with no geological correction. This experiment suggests that: overestimating (underestimating) the difference in density of a rock unit with its true values leads to underestimating (overestimating) its volume, by increasing (decreasing) its overall volume while maintaining its overall shape.

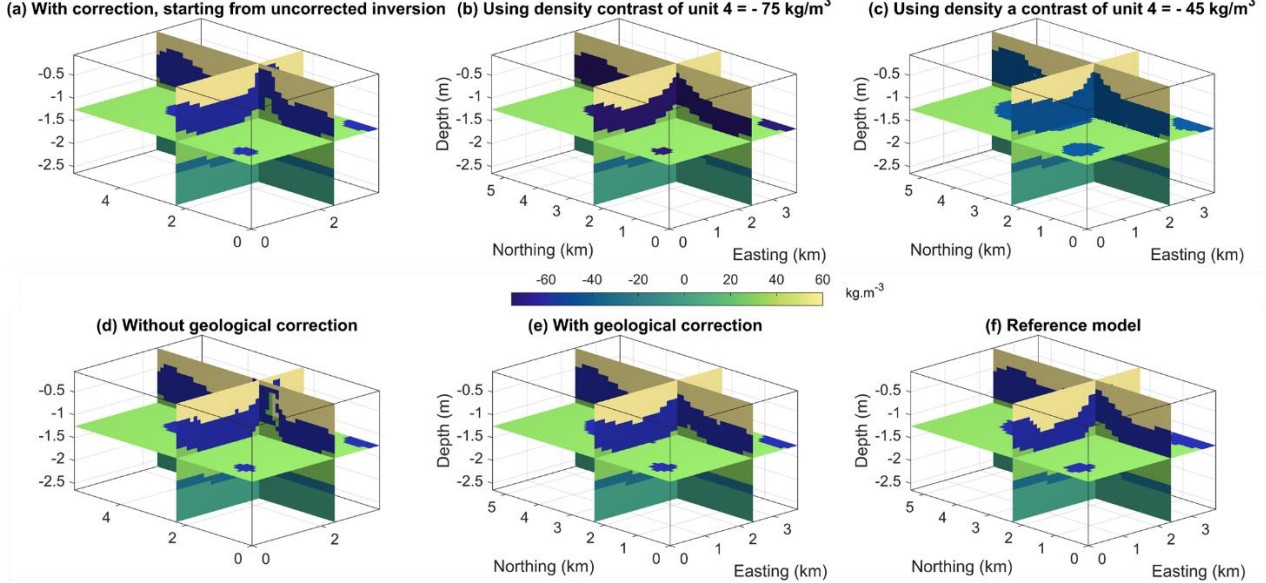

 **Figure 19. Comparison of inversion results with (a) geological correction starting from inverted model obtained without geological correction, as shown in (d); results with geological correction at each iteration with underestimated density contrast (c), overestimated density contrast (d) for unit 4. The case with accurate density contrast for all units unit and the reference model are show in (d) and (e), respectively.**

### 13.6. Appendix 6: Geological data

**Table A 1. Geological data used to build the reference geological model**

| Stratigraphic column | | | Stratigraphic 'younging' vector | | |
|---|---|---|---|---|---|
| Rock unit | Min. thickness | Max. thickness | 0 | √2/2 | √2/2 |
| (a) | -Inf | 0 | Contact between unconformity and stratigraphy | | |
| (b) | 0 km | 2.1 km | X (m) | Y (m) | Z (m) |
| (c) | 2.1 km | 6 km | 0 | -5000 | 0 |
| (d) | 6 km | Inf | Coordinates of vector normal to the fault plane | | |
| Coordinates of vector normal to unconformity | | | X (m) | Y (m) | Z (m) |
| X (m) | Y (m) | Z (m) | 0.5 | -0.5 | -0.5 |
| 0.5 | -0.5 | 0.5 | Fault slip vector | | |
| Fault displacement length (m) | | | X (m) | Y (m) | Z (m) |
| 3.75 km | | | 0 | 1 | 0 |