# Peer review of "automatic implicit geological modelling Integration of in deterministic geophysical inversion"

_EGUsphere, 2023_

## Author Comment (AC1)

**Manuscript**: egusphere-2023-129

**Object**: answer to Anonymous Referee #1, comment posted on 06 Mar 2023

**Citation of original comments**: https://doi.org/10.5194/egusphere-2023-129-RC1

Dear Reviewer,

Thank you for taking the time to review our submission and for your comments. Below you will find our answers in blue.

We appreciate your positive feedback.

Best regards,

The Authors
* * *
After carefully reading and evaluating "Integration of automatic implicit geological modelling in deterministic geophysical inversion", I am impressed with the thoroughness and meticulousness of the research. Good work! The authors have made a substantial contribution to both the geophysics and geology communities with their framework, which can be readily applied to a range of geoscientific problems using gravity data. This includes exploring critical minerals and mapping volcano structures. Overall, I believe this manuscript is of significant importance and quality, and I am pleased to recommend it for publication in Solid Earth, pending a moderate revision.

Thanks for your feedback!

Below please find my major concerns followed by specific comments.

Major concerns:

1. The quality of the geological model is a critical aspect of the framework presented in the manuscript. It is noted that constructing a detailed prior geological model is not commonly practiced, as it requires a significant amount of field geological data and expert knowledge. A reference model that contains incorrect information may guide inversion in the wrong direction, even though the uncertainty may be reduced. The question arises as to whether geophysical data can correct any errors in the reference model. The authors mention that "their approach allows for the inversion to explore a part of the model space that remains within the neighborhood". However, it is unclear how robust this method is. Therefore, further clarification on the efficacy of this approach is warranted, especially in dealing with errors in the reference model.

This is correct. Two aspects can be considered in this question.

The first aspect relates to the impact of poor starting geological models. This has been investigated for the multi-unit geophysical inversion in previous works by Giraud et al. (2021) and Rashidifard et al. (2021), which have shown that, to a certain extent, the type of modelling we propose is robust to errors in the geometry of geological units. RC2 and RC3 raised the issue of using incorrect starting models and we refer to our answer to RC2's first point and RC3's third point for more details. We performed several tests to investigate the impact of a geologically incorrect starting model and found that our method is robust to this to a certain extent (see Figure we added in Section "5.3. Improving the geological realism of a pre-existing model", and "Appendix 4: Robustness to a degenerate starting model with noisy data" and "Appendix 5: Robustness to errors in the density of rock units").

The second aspect relates to conflicts between geological concepts or parameterization and geophysical data. We agree that this calls for further scrutiny. You wrote "The question arises as to whether geophysical data can correct any errors in the reference model." The answer to this question is negative, as some errors that are not incompatible with the geophysical data may persist due to the non-uniqueness of the inverse problem. However, the severity of such errors may be reduced (see the examples we ran with poor starting models). We argue that errors in the geological parameterisation such as the number of rock units or, more importantly, the geological principles that are used, may be deduced. For instance, one may compare the data misfits for the case with geological correction, and without. In the event that the geological parameterisation is affected by such errors, it is likely that the application of a geological correction or using geological prior model term might prevent proper convergence of the inversion.

2. I recommend that the authors provide further elaboration on the term "rock unit" utilized in the manuscript. Although it is mentioned that each unit is associated with a unique range of physical properties and retains constant physical property values during inversion, it is unclear whether a unit corresponds to a single lithology or a mixture of multiple lithologies. Additionally, it is possible for a single lithology to be present in zones that exhibit varying degrees of alteration, mineralization, or mineral assemblages, leading to different physical properties. As a result, further clarifications regarding the definition of rock unit are required to enhance the understanding of the framework presented in the manuscript.

We agree with this comment. What we call a 'unit' is a unit in the petrophysical property contrast sense. Each 'unit is a geological entity that has a petrophysical value (here: density) that is different from the rest. For instance, it is therefore possible for several rock types from the same stratigraphy to fall under the same 'unit' in the modelling approach that we follow.

We clarify this in the methodology section in Section 2.1 by appending the following sentence (underlined text added):

"The method we present relies on the formulation of the model using an implicit model formulation in the form of signed distances to interfaces between rock units. As proposed by Giraud et al. (2021a), this modelling approach considers 'rock units' as one or more rock types characterised by the same physical value (e.g., here, each unit is characterised by a single density value within the modelled area). Each rock unit is modelled by a unique signed-distance scalar field covering the study area. In a study considering $n_r$ rock units of known contrasting physical properties, we consider a set of $n_r$ signed-distance fields $\boldsymbol{\phi} = \{\boldsymbol{\phi}_k, k = 1, \dots, n_r\}$ over $n_m$ model cells corresponding to the distance to the boundaries of rock units."

In the discussion, we added the following in Section 6.2:

"It is therefore possible for several rock types from the same stratigraphy to fall under the same 'unit' in the modelling approach that we follow."

3. Perturbing dipping angle, azimuth, and strike for one or multiple units is a simple way to quantify the uncertainty of the deterministic inversions. Have you done any experimental results regarding uncertainty quantification? I would suggest mentioning UQ in Discussion.

The interpolation algorithm used in LoopStructural minimises the misfit between the constraining data and the regularisation term (smoothing). This means that adding uncorrelated noise to the constraints through a perturbation would introduce a local misfit, the impact of which would be removed, or reduced significantly, by the regularisation. An example for this is presented in Grose et al. (2021) which shows a comparison between radial basis function and discrete interpolation for noisy data points.

Nevertheless, it would be possible to force the perturbed data points to be more strongly weighted in the model to generate different geological models. We argue that perturbing the area in the geological model that is actually constrained by geological observations may not achieve the desired outcome of simulating geological uncertainties. We argue that uncertainties increase away from observations due to the lack of geological knowledge used during interpolation but this is not necessarily captured when perturbing observations. In the scheme of a geophysical inversion, these models might show areas with higher uncertainties around the locations of observations and of structures that are constrained by the geological data and not elsewhere. Ideally, combining geological modelling and geophysical inversions will help geophysics inform the model away from geological data.

We agree that UQ is an important aspect, and it is a topic that we are currently investigating. We are considering perturbations of inversion results to perform null space analysis, but this is not the object of this submission and doing so would make a lengthy paper. Nonetheless,, we have added the following to the discussion (Section 6.1):
"Further considerations of uncertainty may be required to better evaluate and understand inversion results. For instance, the approach of Wei and Sun, 2022, who generate series of inverted models by varying their

deterministic inversions' hyperparameters could be a source of inspiration for uncertainty estimation. Likewise, the scalar field perturbation of Henrion et al., 2010, Clausolles et al. (2023) or Yang et al. (2019) could be transposed to the modelling approach we propose here."

We appended the following to the discussion (Section 6.2):

"Finally, as mentioned in several places in the paper, further considerations on uncertainty are also required to better understand inversion results and find a set of admissible models, both geologically and geophysically."

And also added elements of discussion in Section 6.3 as we detail in our answer to you next point.

4. The assigned density values utilized as a priori information can have a significant impact on the resulting model recovery. Small density values may lead to an overestimation of the rock unit's volume, resulting in a large and less dense body, while large density values may result in the underestimation of the volume corresponding to a small but dense body. To provide a more comprehensive understanding of how prior physical property values affect model reconstruction, it would be valuable to include numerical results reflecting the impact of the assigned density values. This information would enable readers to appreciate the extent of the influence of the a priori information on the inversion outcomes and aid in the interpretation of the model results.

We agree. Generally speaking we can expect (and have observed) that overestimating the density of a rock unit will lead to underestimating its volume, and conversely for an underestimation of density. In other words, the rock unit affected would either inflate or deflate depending on whether the density is underestimated or overestimated. This is illustrated by additional tests we performed using the first synthetic model. The Figure reproduced below was added as an Appendix to the manuscript.

[Figure]

*Figure 1. Comparison of inversion results with (a) geological correction starting from inverted model obtained without geological correction, as shown in (d); results with geological correction at each iteration with underestimated density contrast (c), overestimated density contrast (d) for unit 4. The case with accurate density contrast for all units unit and the reference model are show in (d) and (e), respectively.*

> We have add this Figure in an appendix (Appendix 'Robustness to errors in the density of rock units') along with a paragraph discussing the results.
>
> The importance of prior information such as the number of units or their density has been investigated in a previous work using a similar approach, e.g, Giraud et al. (2021). Nevertheless, we agree that this is important and have added the following text to further discuss the importance of prior models.
>
> We modified the discussion in Section 6.3 as follows (underlined text added):
>
> " Finally, extensions of our method may allow for null-space exploration and the mitigation of some of the limitations identified in the previous subsection. This paper investigates the importance of  geological information in level set inversion. Previous work focusing on level set inversion following an approach similar to ours have investigated the importance of accurate knowledge on the geometry and the number of rock units a priori (Giraud et al., 2021a). Giraud et al., 2021a, and Rashidifard et al., 2021, suggest that inversion is somewhat robust to errors in the starting model geometry and in the petrophysics of the rock units. However, results from Giraud et al., 2021a, using synthetic models suggest that level-set inversion "presents limitations when an important geologic unit is missing from the initial model". To alleviate this, ways to generate the 'birth' of new geological units for inversion to consider geological bodies previously not accounted due to lack of information may be devised. One possibility could be to use the sensitivity of geophysical data to changes in physical property."

Specific comments:

The manuscript is in a good shape, I just have a few minor comments.

Page 2L40: It is recommended that the authors consider citing the work of Wei and Sun (2022) in the manuscript. Wei and Sun quantified the uncertainty of rock units derived from geophysical joint inversion. Including a citation to this work would provide readers with additional context and insights into the understanding of rock units.

*Wei, X. and Sun, J., 2022. 3D probabilistic geology differentiation based on airborne geophysics, mixed L p norm joint inversion, and physical property measurements. Geophysics, 87(4), pp.K19-K33.*

> Thanks for the reference. We have added the following to the discussion (same text the answer to point 3 above about UQ).

Section 6.1:
"Further considerations of uncertainty may be required to better evaluate and understand inversion results. For instance, the approach of Wei and Sun, 2022, who generate series of inverted models by varying their deterministic inversions' hyperparameters could be a source of inspiration for uncertainty estimation. Likewise, the scalar field perturbation of Henrion et al., 2010, Clausolles et al. (2023) or Yang et al. (2019) could be transposed to the modelling approach we propose here."

Section 6.2:
"As mentioned in several places in the paper, further considerations of uncertainty may be required to be able to better understand inversion results."

Section 6.3:

"… constitute a promising area of research for future work, and that uncertainty quantification needs more investigations"

Page 19 Figure 11: I would like to see the observed gravity data in Fig 11 as well, which can tell the mismatch between observed data and simulated data using reference and starting model.

We have added the following underlined text to the manuscript in Section 5.2.1.:

"The gravity data simulated for the reference and starting models are shown in Figure . We consider the data produced by the reference model as the field measurements corresponding to the model we try to recover."

In the caption of Figure 11, we have added the following:

"Figure 11. Gravity data simulated for the reference model (left), which we consider as the field data that invert, and for the starting model (right), which corresponds to the starting point of inversions."

Page 22 Figure 15: It is not clear from the manuscript why Cases 2, 3, and 4 do not converge.

All inversions actually do converge to a stable solution to a similar data misfit of 0.5 mGal. However, they converge to different models, each of which is characterised by a different value for $OC$, $ERR_m$ and $ERR_\phi$. We have added the following clarification (Section 5.2.2):

"This observation is further confirmed by the metrics shown in Figure 12. All four inversions converge to different models, each of which is characterised by a different value for $OC$, $ERR_m$ and $ERR_\phi$."

**References**

Clausolles, N., Collon, P., Irakarama, M., and Caumon, G.: Stochastic velocity modeling for assessment of imaging uncertainty during seismic migration: application to salt bodies, Interpretation, 1–67, https://doi.org/10.1190/int-2022-0071.1, (2023).

Giraud, J., Lindsay, M., and Jessell, M.: Generalization of level-set inversion to an arbitrary number of geologic units in a regularized least-squares framework, GEOPHYSICS, 86, R623–R637, https://doi.org/10.1190/geo2020-0263.1, (2021).

Grose, L., Ailleres, L., Laurent, G., and Jessell, M.: LoopStructural 1.0: time-aware geological modelling, Geosci. Model Dev., 14, 3915–3937, https://doi.org/10.5194/gmd-14-3915-2021, (2021).

Henrion, V., Caumon, G., and Cherpeau, N.: ODSIM: An Object-Distance Simulation Method for Conditioning Complex Natural Structures, Math. Geosci., 42, 911–924, https://doi.org/10.1007/s11004-010-9299-0, (2010).

Rashidifard, M., Giraud, J., Lindsay, M., Jessell, M., and Ogarko, V.: Constraining 3D geometric gravity inversion with a 2D reflection seismic profile using a generalized level set approach: application to the eastern Yilgarn Craton, Solid Earth, 12, 2387–2406, https://doi.org/10.5194/se-12-2387-2021, (2021).

Wei, X. and Sun, J.: 3D probabilistic geology differentiation based on airborne geophysics, mixed Lpnorm joint inversion and physical property measurements, Geophysics, 87, https://doi.org/10.1190/geo2021-0833.1, (2022).

Yang, L., Hyde, D., Grujic, O., Scheidt, C., and Caers, J.: Assessing and visualizing uncertainty of 3D geological surfaces using level sets with stochastic motion, Comput. Geosci., 122, 54–67, https://doi.org/10.1016/j.cageo.2018.10.006, (2019).

---

## Author Comment (AC2)

**Manuscript**: egusphere-2023-129

**Object**: answer to Anonymous Referee #2, comment posted on 27 Mar 2023

**Citation of comment**: https://doi.org/10.5194/egusphere-2023-129-RC2

Dear Reviewer,

Thank you for taking the time to review our submission and for your comments. Below you will find our answers in blue.

Thank you for your encouragements!

The authors
* * *
**Summary**

The inversion of geophysical data is non-unique. This non-uniqueness is often tackled by spatial regularization promoting simple models, where simplicity is often formulated in terms of spatial smoothness and fails to honor geological realism. The authors present a novel framework to honor constraints derived from implicit geological modeling during the deterministic inversion of geophysical data and demonstrate its advantages using different synthetic geological models and synthetically-generated gravity data. The paper is well-written and accompanied by 17 figures of good quality. I recommend the publication of this manuscript subject to minor revisions. Aside from some specific comments below, I have several suggestions for improvement:

- The authors demonstrate their method on a number of synthetic case studies. In all of these, the starting models is relatively close to the true reference model used to create the synthetic data. A more detailed analysis on how the proposed method depends on the starting model would be insightful.

  A similar comment was made by the other reviewers, and we refer to the Appendices mentioned in our general answer and to the text and figure we added at the beginning of Section 5.3 with a starting model that is geologically unrealistic but which fits the gravity data. What we refer to as a "degenerate starting model" is obtained by rotating the original starting model by 180 degrees around the vertical axis.

  In addition, we already ran one test with a starting model that is geologically unrealistic. This is shown in case (5) of the second synthetic model. To make this more complete, we performed a similar exercise with the first synthetic model where we start inversion with geological correction applied to the results of the uncorrected case. We added this in Section 5.3.:

"To simulate this, we first perform inversion with geological correction applied, starting from the uncorrected case in the synthetic example presented in 5.1. The inverted model obtained in this fashion is shown in Figure 15a. The starting model is shown in Figure 15b. The application of geological correction results in the removal of a number of unrealistic features from the uncorrected inversion results used as a starting model. Here, results obtained by application of geological correction to unrealistic inversion results seem to present an intermediate case between the scenario with application of geological correction from the onset of inversion (Figure 15c) and the uncorrected case (Figure 15b). On this premise, if large unrealistic features appear, we recommend to run geological correction from the beginning of inversion instead of as an ad-hoc process. We note that in the transition between Figure 15b and Figure 15a by application of geological correction, all models are geophysical equivalent in that they present similar geophysical data misfit values. This shows that, for this dataset, there exists a continuum of models with discrete density values fitting the geophysical data to a similar level while presenting different degrees of geological realism. In what follows, we investigate this possibility further using the synthetic dataset presented in Sect. 5.2."

[Figure]

*Figure 1. Comparison of inversion results with (a) geological correction starting from inverted model obtain without geological correction, as shown in (b); results with geological correction (c) and reference model (d).*

In addition, we have added the following to the manuscript, in the discussion section:

'Previous work focusing on level set inversion following an approach similar to ours have investigated the importance of accurate knowledge on the geometry and the number of rock units a priori (Giraud et al., 2021a). Giraud et al., 2021a, and Rashidifard et al., 2021, suggest that inversion is somewhat robust to errors in the starting model geometry and in the petrophysics of the rock units.

Nonetheless, relatively small deviations between, scenarios 2 and 5 illustrate that the proposed methodology is not sufficient to address the ill-posedness of the potential field problem. Moreover, results from Giraud et al., 2021a, suggest that level-set inversion "presents limitations when an important geologic unit is missing from the initial model"'

- Inversion models should not under- or overfit the measured data. The very important topic of measurement error and data weights during the inversion should deserve more attention in this manuscript. Was the synthetic data noisified before inversion? If yes, which type of noise was used (e.g., Gaussian White Noise)? How were measurement errors treated in the inversion? Currently, this seems a bit arbitrary (e.g., no data weights in the formulas, inversion stops at 0.5 mGal.).

  We assume that convergence was reached when data misfit reached 0.5 mGal. We chose this value accordingly with values obtained from the literature (Barnes et al., 2011). We have added the following to the manuscript in section 5.2.1 Survey Setup (underlined text):

  "We run inversion corresponding to the four inversion scenarios proposed above. Inversions stop when reaching $ERR_d = 0.5$ mGal, which corresponds to acceptable values for legacy data (Barnes et al., 2011). The gravity data simulated for the reference and starting models are shown in Figure 9. We consider the data produced by the reference model as the field measurements corresponding to the model we try to recover. We assume zero error to test the ability of the method to recover the reference in a perfect data settings (See Appendix 3: Robustness to noise and Appendix 4: Robustness to a degenerate starting model for a more realistic case including data errors)."

  In the methodology section, we have added the following after the description of the cost function (eq. 4 in section 2.2).

  "We write the data misfit term as: $\|\boldsymbol{d}^{obs} - \boldsymbol{d}^{calc}\|_2^2$ instead of $\|\boldsymbol{W_d}(\boldsymbol{d}^{obs} - \boldsymbol{d}^{calc})\|_2^2$ with $\boldsymbol{W_d}$ would be the data (co)variance matrix. In our current implementation, we simplify the problem by assuming that uncertainty (or noise) in the data is isotropic. This is equivalent to assuming an isotropic covariance matrix $\boldsymbol{W_d}$ and adjusting the other hyperparameters accordingly. An obvious extension of the work presented here is to consider a matrix $\boldsymbol{W_d}$ with values that can be set individually for each datum."

  Although we agree that data noise can have an impact in practical studies, we did not add noise to the data, so the data is noise-free. One reason is to test the ability of this new method to recover sensible results in an ideal case. Also, in the tests that JG ran when developing the method, he observed that it was robust to Gaussian random noise. A possible explanation is that Gaussian random noise may have an effect on physical property inversion as it can cause small scale anomalies in the recovered model. The severity of this is reduced in the case of geometrical inversion because the densities remain constant within the different rock units. Consequently, the risk of fitting random noise is reduced with this kind of inversion. However, it could be that longer wavelength correlated noise affect the inversion results, but it is generally the case for many inversion schemes.

> We investigate the impact of noise with the simulation of contaminated data in Appendix 3 and 5, which we refer to for more information, as mentioned in our general answer.

- Discussion of existing literature with similiar motivations: While it is true that the majority of literature either attempts to invert geophysical data using smoothness-constraints or simple geometrical definitions such layer-based parameterizations, there are a few recent examples where a geological modeling engine was used in the inversion of geophysical data. Putting these into context is in my view much more valuable for the readers (and the impact of the presented method) than pointing out differences to conventional approaches such as Thikonov regularization.

  - Güdük, N., de la Varga, M., Kaukolinna, J., & Wellmann, F. (2021). Model-Based Probabilistic Inversion Using Magnetic Data: A Case Study on the Kevitsa Deposit. In Geosciences (Vol. 11, Issue 4, p. 150). MDPI AG. https://doi.org/10.3390/geosciences11040150

  - Liang, Z., Wellmann, F., & Ghattas, O. (2022). Uncertainty quantification of geologic model parameters in 3D gravity inversion by Hessian-informed Markov chain Monte Carlo. In GEOPHYSICS (Vol. 88, Issue 1, pp. G1–G18). Society of Exploration Geophysicists. https://doi.org/10.1190/geo2021-0728.1

    > Thanks for the references. We have added a citation of Liang et al. (2023) in the introduction:
    >
    > "To mitigate this, several solutions may be devised. One possibility, explored recently by Güdük et al. (2021) and Liang et al. (2023), consists in computing the geophysical response directly on geological models".
    >
    > We have added the following to the Discussion (section 6.1):
    >
    > "Finally, the approach presented here belongs to a family of inversion approaches that could be referred to as 'geometry driven'. That is, the main driver for fitting the geophysical data is the geometry of the subsurface model. One of our working assumptions is that geological data used to derive structural geological models are complemented by geophysical data, with the possibility to alter the shape of geological models. Another 'geometry driven' approach, consists in sampling points controlling the interpolated geological models within uncertainty and to integrate their forward geophysical response to the calculation of a posterior distribution (Güdük et al., 2021, Liang et al., 2023). This method is very elegant as it uses a unique geological level set parameterization, but it does not explicitly address the integration of spatial geological data as provided by drillholes or reflection seismic data. In contrast, our method can honour (up to discretization errors) spatial data. The mapping between the geophysical level set $\phi$ and geological level sets $\phi$ makes it challenging to derive sensitivities directly as in Liang et al. (2023), but it probably makes it possible to explore a larger model space by possibly departing from geological acceptable solutions during the non-linear iterations. Although further tests and comparisons between the two approaches are probably needed, this feature could be useful to prevent the optimization from converging to local minima and possibly in the future for stochastic inversions"

To add more context about other techniques pursuing the same goal, we have also added the following in the introduction:

"More recently, surface-based inversion has become practical for seismic and for potential field inversion by using either an explicit formulation for the interface between rock units (Galley et al., 2020) or using the level-sets in the inversion (e.g., Dahlke et al., 2020; Giraud et al., 2021a; Li et al., 2017, 2020; Rashidifard et al., 2021; Zheglova et al., 2018)"

**Specific comments**

- L52: In near-surface geophysics, "petrophysical (joint) inversion" is a widespread term implying the use of petrophysical relations for parameter transformation (e.g., electrical conductivity to water content) within the parameter estimation.

  We have rephrased the sentence that we thought was ambiguous to:

  "In comparison to the direct inversion of  physical properties (i.e., density, electrical resistivity, seismic velocities, etc.), these geometrical inversions …"

- L119 / L120: "regularization" vs. "regularisation". Check the entire mansucript for the consistent user of either British or American English.

  Thanks for picking on this. We have corrected the case of regularization vs regularisation and other such words.

- L132: Is the opening bracket before "see" intended? If yes, make sure it closes.

  Yes it was intended; problem corrected.

- Eq. 6: This seems to be redundant. Maybe psi^prior could be already introduced in equation 4?

  We agree that this is redundant. We have removed the equation. As psi^prior is not referred to later in the text, we just removed it.

- L181: Remove the comma after data

  Done, thanks.

- L199: Mentioning a few alternative geological modeling engines which could be used seems useful here.

  We have added references to GeoModeller and GemPy, SKUA-GOCAD, Petrel and Leapfrog. The modified text is as follows:

  "…be carried out with other implicit geological modelling engines, such as through GeoModeller's application programming interface Calcagno et al. (2008); Guillen et al. (2008), GemPy De La Varga et al. (2019), SKUA-GOCAD (Jayr et al. (2008)), Petrel Souche et al. (2015), or Leapfrog Cowan and Beatson (2002)"

- Eq. 16: Measurement errors seem to be neglected here. Was noise use to make the synthetic experiments more realistic? If yes, a data error estimate should also be used in the inversion. This needs to be clarified.

  We provide an answer to this question in our reply to your general comment above.

- L382: "... the data misfit visible in Fig. 5a and Fig. 5b" is misleading here, since no data misfits are shown. I recommend to rephrase to "... the corresponding data misfit of the inversion results shown in Fig. 5a and Fig. 5b".

  We have rewritten part of the sentence as:

  "… the requirement to reduce the data misfit component of Eq. (4) (the evolution of which is shown in Figure 3a and Figure 3b) leads the inversion to produce …"

- L459: Why 0.5 mGal? Was the data noisified and would this value mean the data was descirbed within its error bounds? The very important topic of data error/weights should deserve a bit more attention in this manuscript.

  We provided an answer to this issue when addressing your general comment above (second bullet point).

- L660: Is the code related to the method presented in this paper part of LoopStructural or is it made available with this manuscript in a separate repository?

  The inversion code is not yet available. We clarity this with the following text:

  "The inversion used here is a prototype under development that will be released in the future." in the data and code availability section.

- L738: The link to the pdf refers to a local filesystem.

  Thanks for spotting this. Corrected.

- L797: The editor's last name is Schmelzbach (last letter h not k), the publisher is Elsevier, and the name of the book series ("Advances in Geophysics", https://www.sciencedirect.com/bookseries/advances-in-geophysics) may be added.

  Thanks for spotting this; we have added the name of the book series.

**References**

Barnes, G. J., Lumley, J. M., Houghton, P. I., and Gleave, R. J.: Comparing gravity and gravity gradient surveys, Geophys. Prospect., 59, 176–187, https://doi.org/10.1111/j.1365-2478.2010.00900.x, (2011).

Calcagno, P., Chilès, J. P., Courrioux, G., and Guillen, A.: Geological modelling from field data and geological knowledge. Part I. Modelling method coupling 3D potential-field interpolation and geological rules, Phys. Earth Planet. Inter., 171, 147–157,

https://doi.org/10.1016/j.pepi.2008.06.013, (2008).

Cowan, J. and Beatson, R.: Rapid Geological Modelling, in: Australian Institute of Geoscientists Bulletin 36, (2002).

Dahlke, T., Biondi, B., and Clapp, R.: Applied 3D salt body reconstruction using shape optimization with level sets, Geophysics, 85, R437–R446, https://doi.org/10.1190/geo2019-0352.1, (2020).

Galley, C. G., Lelièvre, P. G., and Farquharson, C. G.: Geophysical inversion for 3D contact surface geometry, GEOPHYSICS, 85, K27–K45, https://doi.org/10.1190/geo2019-0614.1, (2020).

Giraud, J., Lindsay, M., and Jessell, M.: Generalization of level-set inversion to an arbitrary number of geologic units in a regularized least-squares framework, GEOPHYSICS, 86, R623–R637, https://doi.org/10.1190/geo2020-0263.1, (2021).

Güdük, N., De La Varga, M., Kaukolinna, J., and Wellmann, F.: Model-based probabilistic inversion using magnetic data: A case study on the kevitsa deposit, Geosci., 11, 1–19, https://doi.org/10.3390/geosciences11040150, (2021).

Guillen, A., Calcagno, P., Courrioux, G., Joly, A., and Ledru, P.: Geological modelling from field data and geological knowledge. Part II. Modelling validation using gravity and magnetic data inversion, Phys. Earth Planet. Inter., 171, 158–169, https://doi.org/10.1016/j.pepi.2008.06.014, (2008).

Jayr, S., Gringarten, E., Tertois, A. L., Mallet, J. L., and Dulac, J. C.: The need for a correct geological modelling support: the advent of the UVT-transform, First Break, 26, https://doi.org/10.3997/1365-2397.26.10.28558, (2008).

De La Varga, M., Schaaf, A., and Wellmann, F.: GemPy 1.0: Open-source stochastic geological modeling and inversion, Geosci. Model Dev., 12, 1–32, https://doi.org/10.5194/gmd-12-1-2019, (2019).

Li, W., Lu, W., Qian, J., and Li, Y.: A multiple level-set method for 3D inversion of magnetic data, GEOPHYSICS, 82, J61–J81, https://doi.org/10.1190/geo2016-0530.1, (2017).

Li, W., Qian, J., and Li, Y.: Joint inversion of surface and borehole magnetic data: A level-set approach, GEOPHYSICS, 85, J15–J32, https://doi.org/10.1190/geo2019-0139.1, (2020).

Liang, Z., Wellmann, F., and Ghattas, O.: Uncertainty quantification of geologic model parameters in 3D gravity inversion by Hessian-informed Markov chain Monte Carlo, Geophysics, 88, G1–G18, https://doi.org/10.1190/geo2021-0728.1, (2023).

Rashidifard, M., Giraud, J., Lindsay, M., Jessell, M., and Ogarko, V.: Constraining 3D geometric gravity inversion with a 2D reflection seismic profile using a generalized level set approach: application to the eastern Yilgarn Craton, Solid Earth, 12, 2387–2406, https://doi.org/10.5194/se-12-2387-2021, (2021).

Souche, L., Lepage, F., Laverne, T., and Buchholz, C.: Depositional Space: Construction and Applications to Facies and Petrophysical Property Simulations, in: Day 2 Mon, December 07, 2015, https://doi.org/10.2523/IPTC-18339-MS, (2015).

Wellmann, J. F., de la Varga, M., Murdie, R. E., Gessner, K., and Jessell, M.:

Uncertainty estimation for a geological model of the Sandstone greenstone belt, Western Australia – insights from integrated geological and geophysical inversion in a Bayesian inference framework, Geol. Soc. London, Spec. Publ., SP453.12, https://doi.org/10.1144/SP453.12, (2017).

Zheglova, P., Lelièvre, P. G., and Farquharson, C. G.: Multiple level-set joint inversion of traveltime and gravity data with application to ore delineation: A synthetic study, GEOPHYSICS, 83, R13–R30, https://doi.org/10.1190/geo2016-0675.1, (2018).

---

## Author Comment (AC3)

**Manuscript**: egusphere-2023-129

**Object**: answer to Anonymous Referee #3, 29 Mar 2023

**Citation of original comments**: https://doi.org/10.5194/egusphere-2023-129-RC3

Dear Reviewer,

Thank you for taking the time to review our submission and for your constructive feedback. Below you will find our answers in blue.

Thank you!

The authors
* * *
After reviewing the manuscript titled "Integration of automatic implicit geological modelling in deterministic geophysical inversion," I was thoroughly pleased to see the development of new methodologies that address the problem of non-uniqueness in geophysical inverse problems. The manuscript is well-written and well-illustrated and has clear value that addresses the integration of geological and geophysical data. I look forward to seeing a field application of the proposed methods. Having said that, I do feel a few revisions would improve the quality of the manuscript.

First, equations (2) and (5) both present ways to map from signed-distances to physical properties, so which one is used in which stage of the inversion? The text (L 128 – 130) states that equation (5) is used to update the model, so when is equation (2) used then? Giraud et al. (2021a) suggest instead that equation (2) is used for the model update. Please clarify this, as equations (2) and (5) seem to be in conflict.

> We have brought clarifications to this, which indeed was a bit confusing. Eq. (2) is used to map signed distances to density contrast when the signed distance properties of $\phi$ are initialised from a model of rock units, which is not the case after model update. Therefore, after each model update, we use equation (5) to obtain the new signed distances $\phi$, which we recalculate at each iteration. Equation (2) is used for the calculation of the sensitivity matrix of the forward data to changes in signed distances. We have clarified this, in L100 of the original manuscript, we have modified the sentence introducing Eq. (2):
>
> "In our level-set inversions, $\phi$ is the primary variable inverted for, which constitutes the proxy for a direct link mapping geophysical and geological representations of the subsurface. We map these signed distances to petrophysical properties using:
>
> (2)"

$$m(\boldsymbol{\phi}_1, \dots, \boldsymbol{\phi}_{n_r}) = \sum_{i=1}^{n_r} V_i H(\boldsymbol{\phi}_i) \left[ \prod_{j=1, j \neq i}^{n_r} \left( 1 - H(\boldsymbol{\phi}_j) \right) \right],$$

After the description of the cost function (Eg. 4), we have added the following:

"To solve for $\delta\boldsymbol{\phi}$, we build the system of equations given in Appendix 1, which requires the sensitivity matrix $\boldsymbol{S}^\phi$ of $\boldsymbol{d}^{calc}(\boldsymbol{m}(\boldsymbol{\phi}))$ with respect to changes in $\boldsymbol{\phi}$ using the chain rule:

$$\boldsymbol{S}^\phi = \frac{\partial \boldsymbol{d}^{calc}(\boldsymbol{m}(\boldsymbol{\phi}))}{\partial \boldsymbol{\phi}^k} = \frac{\partial \boldsymbol{d}}{\partial \boldsymbol{m}(\boldsymbol{\phi}^k)} \frac{\partial \boldsymbol{m}(\boldsymbol{\phi}^k)}{\partial \boldsymbol{\phi}^k} = \boldsymbol{S}^m \frac{\partial \boldsymbol{m}(\boldsymbol{\phi}^k)}{\partial \boldsymbol{\phi}^k},$$ (4)

where $\frac{\partial \boldsymbol{m}(\boldsymbol{\phi}^k)}{\partial \boldsymbol{\phi}^k}$ is obtained analytically from Eq. (2-3) (see Giraud et al., 2021a for details)."

Second, I do feel that while the manuscript does make an effort to explain in detail the formulation of the inverse problems, it falls short in explaining how some of the parameters are determined. For example, the parameter "alpha" in equation (12) for the geological correction seems to be crucial in guiding the inversion, yet there is little mention on how "alpha" was determined. Results are shown for alpha = 0 and 0.5, but there is no explanation of how these values were determined. If "alpha" is a hyperparameter, then a sensitivity analysis that tests various values for "alpha" ranging between 0 and 1 would be appropriate.

Yes, the parameter "alpha" is a hyperparameter of the inversion. We did not perform extensive, rigorous testing of values for this parameter as we originally sought to balance the contribution of geological modelling and geophysical inversion in the search direction and a value of 0.5 gives a similar importance to both terms. We proceeded somewhat naively from manual fine tuning of alpha with values in the vicinity of 0.5. A value of 0.5 provides satisfactory results, which, at the time, did not warrant the need to perform any more advanced analysis such as, for example, the rigorous plot of an L-curve.

While adding a rigorous analysis of the importance of alpha would be interesting, we prefer to maintain the manuscript focussed and to its current length as much as possible. We added a couple of sentences explain the rationale behind a value of 0.5 in Sect. 5.2.1. (L408-410 of the updated manuscript):

"The value of α = 0.5 was chosen without a rigorous analysis of its impact on the results. A naïve trial and error approach revealed that a value of 0.5 effectively 'corrected' the course taken by un-corrected geophysical inversion and prevented the appearance of artefacts."

Third, there is little mention of the robustness of the inverted models. Granted, this is a synthetic study, but how robust are some of the features that are consistently seen in the inverted models? Would changing the values of some parameters in the inversion result in different models, or would the features appear again (indicating some degree of robustness in the model)? This is also related to the fact that the method seems to be dependent on the starting model. How do the results change if the starting model changes? Ideally the proposed methods would give consistent results even when the

parameters and starting model are changed. I would like to see some form of robustness test of the proposed methodologies.

> Tests using starting models far from the reference models have been performed by Giraud et al. (2021), who tested starting models with a missing unit in the starting model and with erroneous density values. Rashidifard et al. (2021) also performed tests on synthetic data with starting models having a geometry that differs from the reference model. The issue you raise here was also raised by the other two reviewers and therefore this answer bear similarities with what we answered to RC1 and RC2. We ran tests using the first synthetic model:
>
> - inversion with geological correction starting from the uncorrected case (added to 'Section 5.3. Improving the geological realism of a pre-existing model'.
>
> - Inversion with under and over-estimated densities, added to 'Appendix 5: Appendix 5: Robustness to errors in the density of rock units'
>
> We refer you to our answer to point 4 of RC1. Information about additional runs is now provided at the end Section 5.1.2. (see general answer).
>
> While the robustness of level set inversion to noise in the data was shown by previous works cited in Introduction, an additional test with unrealistically high noise contamination is provided in Appendix 3 and 5.
>
> We do not paste the associated appendix here not to dilute the focus of our answer. This new appendix is in the updated manuscript (Appendix 3 and Appendix 5).
>
> We have also added the following to the manuscript, in the discussion section (Sect. 6.3):
>
> "Previous work focusing on level set inversion following an approach similar to ours have investigated the importance of accurate knowledge on the geometry and the number of rock units a priori (Giraud et al., 2021a). Giraud et al., 2021a, and Rashidifard et al., 2021, suggest that inversion is somewhat robust to errors in the starting model geometry and in the petrophysics of the rock units. Nonetheless, relatively small deviations between, scenarios 2 and 5 illustrate that the proposed methodology is not sufficient to address the ill-posedness of the potential field problem. Moreover, results from Giraud et al., 2021a, suggest that level-set inversion "presents limitations when an important geologic unit is missing from the initial model". To alleviate this, ways to generate the 'birth' of new geological units for inversion to consider geological bodies previously not accounted due to lack of information may be devised. One possibility could be to use the sensitivity of geophysical data to changes in physical property"
>
> In addition, one test with a starting model that is geologically unrealistic is shown in case (5) of synthetic model 2.
>
> We have also added supplementary material as mentioned in our general answer.

Fourth, I would like to see the simulated synthetic data (i.e. gravity anomaly maps) and the predicted data from the inversions, as well as a data residual map. The data residual

map would ideally show random patterns. A brief explanation on how the data was simulated would also be appropriate. Was the gravity data treated with random noise, or was it simply forward computed from the models? If the gravity data was treated with noise, how robust are the proposed methods to increasing levels of noise? In a field study, the observables would be gravity data maps and any additional geological information that may constrain the models. Yet, the manuscript does not address sufficiently in depth the data itself.

The issue of noise was also raised by RC2. We use elements from our answer to RC2 here.

In the second synthetic test, we assume that convergence was reached when data misfit reached 0.5 mGal. We chose this value accordingly with values obtained from the literature Barnes et al. (2011). We have added the following to the manuscript in section 5.2.1 Survey Setup (underlined text):

"We run inversion corresponding to the four inversion scenarios proposed above. Inversions stop when reaching $ERR_d = 0.5$ mGal, which corresponds to values accepted for legacy data Barnes et al. (2011)"

In the methodology section, we have added the following after the description of the cost function (eg. 4 in section 2.2).

"We write the data misfit term as: $\|\boldsymbol{d}^{obs} - \boldsymbol{d}^{calc}\|_2^2$ instead of $\|\boldsymbol{W_d}(\boldsymbol{d}^{obs} - \boldsymbol{d}^{calc})\|_2^2$ with $\boldsymbol{W_d}$ would the data (co)variance matrix. In our current implementation, we simplify the problem by assuming that uncertainty (or noise) in the data is isotropic. This is equivalent to assuming an isotropic covariance matrix $\boldsymbol{W_d}$ and adjusting the other hyperparameters accordingly. An obvious extension of the work presented here is to consider a matrix $\boldsymbol{W_d}$ with values that can be set individually for each datum."

In addition, in the tests that JG ran when developing the method, he observed that it showed good robustness to random noise. A possible explanation is that random noise may have an effect on physical property inversion as it can cause small scale anomalies in the recovered model. The severity of this is reduced in the case of geometrical inversion because the densities remain constant within the different rock units. Consequently, the risk of fitting random noise is reduced with this kind of inversion. However, it could be that longer wavelength correlated noise affect the inversion results, but is generally the case for many inversion schemes. As an example, we refer to the test with noisy data mentioned above and in the general answer (see Appendices 3 and 5) and also refer to the text added in relation to our answer to your third point.

Fifth, this is somewhat addressed in the last case (5) of the second synthetic study, but what happens if a geological correction is applied to the inverted model from case (1)? This is also related to what is mentioned in L 57 – 60, where an a posteriori ad hoc process could also ensure geological plausibility of an inverted model. Could it be possible that applying a geological correction after geophysical inversion (not every iteration) gives similar results? Then, an argument could be made that the cumulative geological corrections applied at each iteration could be summed into one single geological correction applied at the end of the geophysical inversion. Or perhaps a

geological correction is not necessary at every iteration. A plot showing the evolution of the geological correction with every iteration would be appropriate here.

We break this point into its main logical parts and answer separately:

*"Fifth, this is somewhat addressed in the last case (5) of the second synthetic study, but what happens if a geological correction is applied to the inverted model from case (1)? Fifth, this is somewhat addressed in the last case (5) of the second synthetic study, but what happens if a geological correction is applied to the inverted model from case (1)? This is also related to what is mentioned in L 57 – 60, where an a posteriori ad hoc process could also ensure geological plausibility of an inverted model. Could it be possible that applying a geological correction after geophysical inversion (not every iteration) gives similar results?"*

Similar to the second synthetic case, if we apply a geological correction term to the first synthetic, starting from the uncorrected case, it lead to a model misfit that is intermediate between the case with geological correction applied from the beginning of inversion and no correction at all. We point out that in the case without correction, the geological realism of the inversion result without geological correction is seriously compromised. This makes the geological correction quite substantial as compared to the corrections made during the iterative process.

There is the risk that once starting from a geologically unrealistic model, the geological correction term might propose the closest geological model, which could be from a family of geological model different from the true model.

We have run inversions with geological correction with a starting model equal to the inversion results of the uncorrected case. The result is shown below (Figure 1 in this document and Figure 13 in the new version of the manuscript):

[Figure]

*Figure 1. Comparison of inversion results with (a) geological correction starting from inverted model obtain without geological correction, as shown in (b); results with geological correction (c) and reference model (d).*

While there are some obvious improvements, some unrealistic features remain. This indicates that in this case, the inversion starts from one local minimum of the geophysical data cost function and ends up to another one, less geologically unfavourable, that is different from the case where geological correction is applied from the beginning. We note that some unrealistic thickness variations for Unit 3 (dark blue unit). On this premise, if large unrealistic features appear, we recommend to run geological correction from the beginning of inversion.

We have added the following prose to describe and interpret this example:

"To with the level set method without geological correction as obtained in Section 5.1 (Figure 16a), and run inversion with geological correction applied. The inverted model obtained in this fashion is shown in Figure 16b. The application of geological correction manages to remove a number of unrealistic features present in the starting model. However, the effect of the geological correction seems visually smaller than in the application of geological correction from the onset of inversion (Figure 16c). On this premise, if large unrealistic features appear, we recommend to run geological correction from the beginning of inversion instead of as an ad-hoc process. We note that in the transition between Figure 16a and Figure 16b by application of geological correction, all models are geophysically equivalent or nearly equivalent in that they present similar geophysical data misfit values. This suggests that, for this dataset, a continuum of models exists fitting the geophysical data to a similar level while presenting different degrees of geological realism. In what follows, we investigate this possibility further using the synthetic dataset presented in Sect. 5.2."

*"Then, an argument could be made that the cumulative geological corrections applied at each iteration could be summed into one single geological correction applied at the end of the geophysical inversion. Or perhaps a geological correction is not necessary at every iteration."*

This would be the ideal case and might work in theory but it is unfortunately not so. The geological correction is non linear and the geological modelling problem is also non unique. At any iteration in the inversion, geological correction calculates the geological image of the current model. It uses it to adjust the search direction of geophysical inversion so that the model evolves along a path made up of a series of geologically realistic models (or, at least, more realistic than without correction). Consequently, it is possible that without geological correction, the inversion converges to a model for whose geological image differs from that obtained with geological correction applied throughout the inversion. In such case, applying geological correction to the inverse model would result in a model that might be relatively close, yet different, from the model obtained with correction applied from the beginning.

We note that the paragraph above holds also for the use of a prior geological modelling term.

*"A plot showing the evolution of the geological correction with every iteration would be appropriate here"*

We agree that showing the evolution can be useful, but we decided not to add it to the paper for length reasons. The GIF image we added as supplementary

material gives an qualitative indication of the importance of the evolution of the geological correction at each iteration.

Following are a few specific/technical comments.

L 38: Presumably "units" refer to rock units. Either add "rock" in front of "units", or define what "units" are in this context

We have added "rock" before "units".

L 45: When the parenthetical phrase is removed, the sentence reads "representing equal to zero." I suggest changing "equal" to "equality"

We have change the phrasing to "In the implicit boundary representation, the units' boundaries correspond to the zero iso-value of the implicit functions representing the signed distance to interfaces."

L 46: Change "inverts" to "invert"

Thanks.

L 51 – 53: The sentence "In comparison … potential field data." makes it sound like petrophysical inversions are not geologically meaningful. Suggest rephrasing or clarify what is meant by geologically meaningful

We replaced meaningful by "In comparison to the direct inversion of physical properties (i.e., density, electrical resistivity, seismic velocities, etc.), these geometrical inversions present a direct pathway to obtaining geologically realistic outcomes from the inversion of potential field data."

We replaced 'meaningful' by 'realistic' in that a model can be readily applicable geologically or directly fed back to a geological modelling engine.

L 103: It is unclear what "is" refers to. Presumably it's the "tau" parameter

Yes this is correct. We have corrected it.

L 149: Add "a" between "as" and "prior"

Done.

L 220: I question the value of including Figure 1. Adding illustrations would add value to Figure 1, and would also elucidate the proposed workflow. I would even suggest a possibility of combining with Figure 2. But as it stands, Figure 1 only contains text and adds little value to the manuscript

Thank you for the suggestion. We took this opportunity to shorten the manuscript and deleted Figure 1. We moved Figure 2 (now Figure 1 in the manuscript) higher up in the text in 3.2.1.

L 316: Is it "qualitative" or "quantitative"?

It is quantitative.

L 362: The sentence "Figure 5a and Figure 5b … starting data misfit" is awkward. Is Figure 5 meant as a reference to the sentence? What is meant by a "strong" data misfit?

Thanks. This sentence does not bring much information so we have removed it.

L 370: In Figure 5, the gravity anomaly maps on the top make it difficult to see the reference and starting models. I would suggest presenting the gravity anomaly maps as a separate figure

We have modified the Figure so that the model is more visible and the data less transparent (Figure 2):

[Figure]

*Figure 2.*

L 412, 515, 655: Either define 'geologify' or rephrase to something along the lines of "ensuring geological realism"

We have added the underlined text: "to use geological correction a posteriori to ensure geological realism (to 'geologify')"

L 426: Similar to above comment, define "younging" or rephrase to something like "direction towards younger strata"

We have defined "younging" and added the following in 5.2.1.: "indicating the direction towards younger strata (later referred to as the 'younging' direction)"

L 431: Add "are" between "area" and "shown"

Done.

L 509: What is the sentence "This indicate." referring to?

> 'This' refers to our analysis of adjacency matrices. We replace "this indicates" by "The comparison of adjacency matrices indicates"

L 511: Why is 'improves' in quotations?

> We used quotations because saying that a model is improved is somewhat suggestive. We have replaced this by "greatly increases the geological realism of the final model".

L 534: Is reference to Figure 3 missing, or what is flow (1) referring to?

> Yes the reference to Figure 3 was missing. Thanks, we have corrected it.

L 539: Did you mean "increase" in OC?

> Yes, together with a reduction of the model misfit. We have corrected the sentence as:
> "… which shows an overall increase in OC and a decrease in model misfit"

L 562 – 563: Sentence "As geophysical inversion … solution space." is not clear. Are you saying that geophysical inversions can be used to explore different regions of the solution space?

> Yes. We have rephrased the sentence as:
> "As geophysical inversion may be sensitive to features geological modelling has little to no sensitivity to, geophysical inversion can be used to explore different regions of the solution space"

L 603: Change "scenarii" to "scenarios"

> Done

L 627: Add "of" after "birth"

> Done

L 641: Remove "to" between "remediate" and "some"

> Done

---

## Author Response (AR1)

Dear Reviewers and Editor(s),

Thank you for taking the time to review the manuscript and for managing the review process. Our detailed answers to the reviewers are provided in the interactive discussion.

We noticed that some comments from Reviewer 1, 2, and 3 raised similar issues. We gave specific attention to these issues and ran additional modelling to answer to the different points made by the reviewers.

In particular, we investigated:

- The influence of noise in the data (Appendices 3 and 5),
- The robustness of the inverted models to a poor starting model (Appendices 4 and 5),
- The influence of noise in the data with a poor starting model (appendix 5).

Information about this was added in Section 5.1.2:

"While analysing the influence of inaccurate knowledge of densities in detail is beyond the scope of this paper, it remains important to ensure that inversions are robust to small errors in density. For this, we refer the reader to Appendix 5 where we simulate errors in the knowledge of unit 4. Previous works using level inversion have investigated the importance of the starting model in the uncorrected case and assume that their conclusions hold. Likewise, we assume robustness of level set inversion to noise in the data as it was shown by previous works cited in Sect. 1. To confirm this and for completeness, we performed additional tests, using:

- data contaminated with noise relatively high compared to the amplitude of the uncontaminated data (Appendix 3: Robustness to noise);
- a degenerate starting model and data contaminated with noise (Appendix 4: Robustness to a degenerate starting model)
- a starting model affected by errors in the density of rocks units (Appendix 5: Robustness to errors in the density of rock units).

A detailed analysis of these tests is beyond the scope of this paper and we refer the reader to these Appendices for more detail. In the remainder of this article, we assume that our approach is robust to random noise and inaccurate starting models."

In addition to the Appendices, we added one Figure in the body of the manuscript text, in Section "5.3. Improving the geological realism of a pre-existing model" to investigate the case where the starting model fits the geophysical data but is geologically inconsistent. Moreover, JG released unpublished material relating to Giraud et al. (2021) to enrich the discussion around the importance of prior information and the potential impact of noise onto level set inversion (Giraud, 2023). To complement the written material, we prepared an animated GIF showing the evolution of the inverted model together with geological inconsistences when inversion with geological correction starts from the uncorrected case, for the first synthetic case (Giraud and Caumon, 2023).

Last, we have also proceeded with minor rephrasing in a few places of the manuscript when it was clarifying the meaning or improving the flow.

Best regards,

The Authors

References

Giraud, J.: Synthetic tests: unconstrained multiple level set inversions with errors in the starting model and noise in the data, 9 pp., https://doi.org/10.5281/zenodo.7919381, (2023).

Giraud, J. and Caumon, G.: Evolution of model and geological inconsistencies during inversion, , 1, https://doi.org/10.5281/zenodo.7920886, (2023).

Giraud, J., Lindsay, M., and Jessell, M.: Generalization of level-set inversion to an arbitrary number of geologic units in a regularized least-squares framework, GEOPHYSICS, 86, R623–R637, https://doi.org/10.1190/geo2020-0263.1, (2021).